# DIVERSIFY AND DISAMBIGUATE:
## OUT-OF-DISTRIBUTION ROBUSTNESS VIA DISAGREEMENT

**Yoonho Lee**[*]
Stanford University

**Huaxiu Yao**
Stanford University

**Chelsea Finn**
Stanford University

## ABSTRACT

Real-world machine learning problems often exhibit shifts between the source and target distributions, in which source data does not fully convey the desired behavior on target inputs. Different functions that achieve near-perfect source accuracy can make differing predictions on test inputs, and such ambiguity makes robustness to distribution shifts challenging. We propose DivDis, a simple two-stage framework for identifying and resolving ambiguity in data. DivDis first learns a diverse set of hypotheses that achieve low source loss but make differing predictions on target inputs. We then disambiguate by selecting one of the discovered functions using additional information, for example, a small number of target labels. Our experimental evaluation shows improved performance in subpopulation shift and domain generalization settings, demonstrating that DivDis can scalably adapt to distribution shifts in image and text classification benchmarks.

## 1 INTRODUCTION

Datasets are often *underspecified*: multiple plausible hypotheses each describe the data equally well (D'Amour et al., 2020), and the data offers no further evidence to prefer one over another. Despite such ambiguity, machine learning models typically choose only one of the possible explanations of given data. Such choices can be suboptimal, causing these models to fail when the data distribution is shifted, as common in real-world applications. For example, examination of a chest X-ray dataset (Oakden-Rayner et al., 2020) has shown that many images of patients with pneumothorax include a thin drain used for treating the disease. A standard classifier trained on this dataset can erroneously identify such drains as a predictive feature of the disease, exhibiting degraded accuracy on the intended distribution of patients not yet being treated. To not suffer from such failures, it is desirable to have a model that can discover a diverse collection of alternate plausible hypotheses.

The standard empirical risk minimization (Vapnik, 1992, ERM) paradigm performs poorly on underspecified data, because ERM tends to select the solution based on the most salient features without considering alternatives (Geirhos et al., 2020; Shah et al., 2020; Scimeca et al., 2021). This simplicity bias occurs even when training an ensemble (Hansen & Salamon, 1990; Lakshminarayanan et al., 2017) because each model is still biased towards simple functions. While many recent methods (Ganin et al., 2016; Sagawa et al., 2020; Liu et al., 2021) improve robustness in distribution shift settings, we find that they fail on data with more severe underspecification. This is because, similarly to ERM, these methods only consider a single solution even in situations where multiple explanations exist.

We propose Diversify and Disambiguate (DivDis), a two-stage framework for learning from underspecified data. Our key idea is to learn a collection of diverse functions that are consistent with the training data but make differing predictions on unlabeled test datapoints. DivDis operates as follows. We train a neural network consisting of a shared backbone feature extractor with multiple heads, each representing a different function. As in regular training, each head is trained to predict labels for training data, but the heads are additionally encouraged to represent different functions from each other. More specifically, the heads are trained to make disagreeing predictions on a separate unlabeled dataset from the test distribution, a setting close to transductive learning. At test time, we select one member of the diversified functions by querying labels for the datapoints most informative for disambiguation. We visually summarize this framework in Fig. 1. DivDis is designed for

---

[*]Email: yoonho@cs.stanford.edu. Code is available at https://github.com/yoonholee/DivDis.

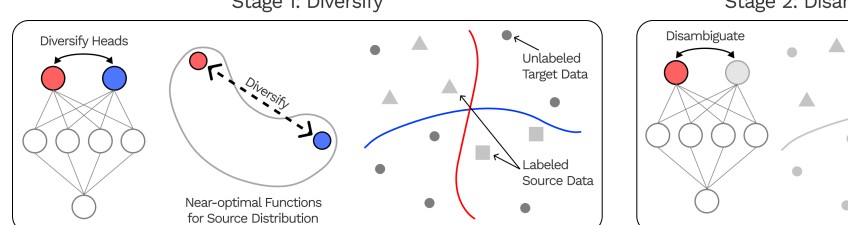

Figure 1: Our two-stage framework for learning from underspecified data. In the DIVERSIFY stage, we train each head in a multi-headed neural network to accurately predict the labels of source data while also outputting differing predictions for unlabeled target data. In the DISAMBIGUATE stage, we choose one of the heads by observing labels for an informative subset of the target data.

scenarios with underspecified data and distribution shift, and its heads will not yield a set of diverse functions in settings where only one function can achieve low training loss.

We evaluate DivDis in several settings in which underspecification limits the performance of prior methods, such as standard subpopulation shift benchmarks (Sagawa et al., 2020) or the large-scale CXR and Camelyon17 datasets (Wang et al., 2017; Sagawa et al., 2022). DivDis achieves an over 15% improvement in worst-group accuracy on the Waterbirds task when tuning hyperparameters without any spurious attribute annotations, and outperforms existing semi-supervised methods on the Camelyon17 task. We also consider challenging problem settings in which labels are *completely correlated* with spurious attributes, so a classifier based on the spurious feature can achieve zero loss. In these completely correlated settings, our experiments find that DivDis is substantially more sample-efficient: DivDis with 4 target domain labels outperforms two fine-tuning methods that use 128 labels.

## 2    LEARNING FROM UNDERSPECIFIED DATA

We consider a supervised learning setting in which we train a model $f$ that takes input $x \in \mathcal{X}$ and predicts its corresponding label $y \in \mathcal{Y}$. We train $f$ with a labeled dataset $\mathcal{D}_S = \{(x_1, y_1), (x_2, y_2), \ldots\}$ drawn from data distribution $p_S(x, y)$. The model $f$ is selected from hypothesis class $f \in \mathcal{F}$ by approximately minimizing the predictive risk $\mathbb{E}_{p_S(x,y)}[\ell(f(x), y)]$ on the data distribution. The model $f$ is evaluated via its predictive risk on held-out samples from $p_S(x, y)$. Standard procedures such as regularization and cross-validation encourage such generalization.

However, even if a function $f$ generalizes to unseen data sampled from the same distribution $p_S(x, y)$, performance often deteriorates in distribution shift conditions, when we evaluate on *target data* sampled from a different distribution $p_T(x, y)$. In many distribution shift scenarios (Koh et al., 2021), the overall data distribution can be modeled as a mixture of domains, where each domain $d \in \mathbb{D}$ corresponds to a fixed data distribution $p_d(x, y)$. In this paper, we specifically consider a subpopulation shift setting, where the source and target distributions are different mixtures of the same underlying domains: $p_S = \sum_{d \in \mathbb{D}} w_d^S p_d$ and $p_T = \sum_{d \in \mathbb{D}} w_d^T p_d$, where $\{w_d^S\}_{d \in \mathbb{D}} \neq \{w_d^T\}_{d \in \mathbb{D}}$.

Conditions like subpopulation shift can be inherently underspecified because the generative process underlying the data distribution, i.e. the domains and coefficients, has so many possibilities. We formalize this intuition through a notion of near-optimal sets of hypotheses. We define the $\varepsilon$-optimal set for a data distribution as follows:

**Definition 1 ($\varepsilon$-optimal set).** *Let $p(x, y)$ be a distribution over $\mathcal{X} \times \mathcal{Y}$, and $\mathcal{F}$ a set of predictors $f : \mathcal{X} \to \mathcal{Y}$. Let $\mathcal{L}_p : \mathcal{F} \to \mathbb{R}$ be the risk with respect to $p(x, y)$. The $\varepsilon$-optimal set with respect to $\mathcal{F}$ at level $\varepsilon \geq 0$ is defined as*

$$\mathcal{F}^{\varepsilon} = \{f \in \mathcal{F} | \mathcal{L}_p(f) \leq \varepsilon\}. \tag{1}$$

Put differently, the $\varepsilon$-optimal set consists of all functions that generalize within the distribution $p(x, y)$. The constant $\varepsilon$ controls the degree of generalization, and we consider small $\varepsilon$ here onwards.

Note that a model's predictions on samples from $p_S(x, y)$—whether $\mathcal{D}_S$ or a held-out validation set—cannot be used to distinguish between different near-optimal functions with respect to $p_S(x, y)$. This is because by definition, the predictions of any two models $f_1, f_2 \in \mathcal{F}^{\varepsilon}$ are nearly identical on $p_S(x, y)$ for small $\varepsilon$. Based on source data alone, we have insufficient reason to prefer any member of $\mathcal{F}^{\varepsilon}$ over another. Our state of belief should therefore cover $\mathcal{F}^{\varepsilon}$ as comprehensively as possible, putting nonzero weight on many functions that embody different inductive biases. This reasoning

is consistent with existing principles for reasoning under uncertainty such as the maximum entropy principle (Keynes, 1921; Jaynes, 1957): in the absence of complete knowledge, one's beliefs should be appropriately spread across all possibilities that are consistent with the available information.

Unfortunately, these principles are notably difficult to implement in practice because of the size and dimensionality of $\mathcal{F}^{\varepsilon}$. We thus incorporate a mild additional assumption into our problem statement that substantially reduces the set of solutions to consider. We use an *unlabeled target dataset* $\mathcal{D}_{\mathrm{T}}$ sampled from $p_{\mathrm{T}}(x)$. The functions inside $\mathcal{F}^{\varepsilon}$ can be compared based on how their predictions differ on $\mathcal{D}_{\mathrm{T}}$. This simplifies the initial infinite-dimensional problem of comparing functions to one of comparing finite sets of predictions, significantly reducing the scope of search. The target set $\mathcal{D}_{\mathrm{T}}$ can also be seen as specifying the directions of functional variation that are most important to us.

We formalize the underspecification with respect to predictions on unlabeled target data through the following variant of $\varepsilon$-optimal sets:

**Definition 2** ($\mathcal{D}\varepsilon$-optimal set). *Let $\mathcal{D} = \{x_1, \ldots, x_n\}$ be an unlabeled dataset, $p(x, y)$ a data distribution, and $\mathcal{F}$ a hypothesis class. Let $\mathcal{F}^{\varepsilon}$ be the $\varepsilon$-optimal set with respect to $p(x, y)$ and $\mathcal{F}$. The $\mathcal{D}\varepsilon$-optimal set $\mathcal{F}^{\mathcal{D}\varepsilon}$ is the set of equivalence classes of $\mathcal{F}^{\varepsilon}$ defined by the following relation $\underset{\mathcal{D}}{\sim}$ between two functions:*

$$f \underset{\mathcal{D}}{\sim} g \quad if \quad f(x) = g(x) \quad \forall x \in \mathcal{D}. \tag{2}$$

Compared to $\mathcal{F}^{\varepsilon}$, the dataset-dependent set $\mathcal{F}^{\mathcal{D}\varepsilon}$ is typically much smaller and easier to manipulate, since it is defined through predictions ($\in \mathcal{Y}$) rather than raw functions.

**Problem statement.** To summarize, we use a labeled *source dataset* $\mathcal{D}_{\mathrm{S}} = \{(x_1, y_1), \ldots\}$ along with an unlabeled *target dataset* $\mathcal{D}_{\mathrm{T}} = \{x_1^t, \ldots\}$. We assume there is subpopulation shift between the two datasets, and our goal is to find a function that performs well in the target distribution. Such a function will lie inside the near-optimal set $\mathcal{F}^{\varepsilon}$ of $p_{\mathrm{S}}(x, y)$, and the learner leverages $\mathcal{D}_{\mathrm{T}}$ to find diverse functions inside this set.

## 3 DIVERSIFY AND DISAMBIGUATE

We now describe Diversify and Disambiguate (DivDis), a two-stage framework for learning from underspecified data. We first describe the general framework (Sec. 3.1), and then our specific implementation of the two DIVERSIFY (Sec. 3.2) and DISAMBIGUATE (Sec. 3.3) stages.

### 3.1 GENERAL FRAMEWORK

As a running example to motivate our algorithm, consider an underspecified cow-camel image classification task in which the source data includes images of cows with grass backgrounds and camels with sand backgrounds. We can imagine two completely different classifiers each achieving perfect accuracy in the source distribution: one that classifies by animal and the other by background. After identifying both possible functions, we can resolve the ambiguity by observing the label of a single image of a cow in the desert.

With this motivation, DivDis aims to first find a set of diverse functions and then choose the best member of this set with minimal supervision. The DivDis framework consists of two stages. In the first stage, we DIVERSIFY by training a finite set of functions that together approximate the $\mathcal{D}\varepsilon$-optimal set for target data $\mathcal{D}_{\mathrm{T}}$. This stage uses both the source and target datasets for training. The source data ensures that all functions achieve low predictive loss on $p_{\mathrm{S}}(x, y)$, while the target data reveals whether or not the functions rely on different predictive fea-

---

**Algorithm 1** DivDis training

**Input:** Source data $\mathcal{D}_{\mathrm{S}}$, Target data $\mathcal{D}_{\mathrm{T}}$, Initial parameters $\theta_0$, Heads $N$, Weights $\lambda_1, \lambda_2$

**Stage one: DIVERSIFY** $\qquad\qquad$ ▷ Sec. 3.2
**while** Not converged **do**
$\quad (X_{\mathrm{S}}, Y_{\mathrm{S}}) \sim \mathcal{D}_{\mathrm{S}}, X_{\mathrm{T}} \sim \mathcal{D}_{\mathrm{T}}$
$\quad \ell_{\mathrm{xent}} \leftarrow \mathcal{L}_{\mathrm{xent}}(\theta; X_{\mathrm{S}}, Y_{\mathrm{S}})$
$\quad \ell_{\mathrm{MI}} \leftarrow \mathcal{L}_{\mathrm{MI}}(\theta; X_{\mathrm{T}})$ $\qquad\qquad$ ▷ (3)
$\quad \ell_{\mathrm{reg}} \leftarrow \mathcal{L}_{\mathrm{reg}}(\theta; X_{\mathrm{T}}, X_{\mathrm{S}})$ $\qquad$ ▷ (4)
$\quad \theta \leftarrow \theta - \alpha \nabla_\theta (\ell_{\mathrm{xent}} + \lambda_1 \ell_{\mathrm{MI}} + \lambda_2 \ell_{\mathrm{reg}})$ ▷ (5)

**Stage two: DISAMBIGUATE** $\qquad\quad$ ▷ Sec. 3.3
$\mathcal{D}_{\mathrm{T}}^* \leftarrow$ Top $m$ datapoints w.r.t. disagreement
**for** Head $i = 1..N$ **do**
$\quad \mathrm{Acc}_i \leftarrow$ Accuracy of head $i$ on $\mathcal{D}_{\mathrm{T}}^*$
Return $\arg\max_i \mathrm{Acc}_i$

---

tures. In the second stage, we DISAMBIGUATE by choosing the best member among this set of functions, for example, by observing the label of a target datapoint for which the heads disagree on.

## 3.2 DIVERSIFY : TRAIN DISAGREEING HEADS

As previously described, the DIVERSIFY stage learns a diverse collection of functions by comparing predictions for the target set while minimizing training error. In our hypothetical cow-camel task, diversifying predictions in this way will produce functions that disagree on ambiguous datapoints like cows in the desert. Such disagreement can cause one head to become an animal classifier and another to be a background classifier.

We represent and train multiple functions using a multi-headed neural network with $N$ heads. For an input datapoint $x$, we denote the prediction of head $i$ as $f_i(x) = \widehat{y}_i$. We ensure that each head achieves low predictive risk on the source domain by minimizing the cross-entropy loss for each head $\mathcal{L}_{\text{xent}}(f_i) = \mathbb{E}_{x,y \sim \mathcal{D}_\text{S}}[l(f_i(x), y)]$.

Ideally, each function would rely on different predictive features in the input. While the most straightforward solution is to enforce differing predictions from each function, this does not necessarily indicate that the two functions rely on different features. As an extreme example, consider a function $f$ for a binary classification problem and its "adversary" $\bar{f}$ which outputs the exact opposite of $f$ on the target dataset. Even though $f$ and $\bar{f}$ disagree completely on the target distribution, they rely on the same input features.

We train each pair of heads to produce predictions that are close to being statistically independent from each other. Independence of prediction values directly implies disagreement in predictions: for example, mutual information is maximized when two prediction heads completely agree with each other. Concretely, we minimize the mutual information between each pair of predictions:

$$\mathcal{L}_{\text{MI}}(f_i, f_j) = D_{\text{KL}}\left(p(\widehat{y}_i, \widehat{y}_j) \,||\, p(\widehat{y}_i) \otimes p(\widehat{y}_j)\right), \tag{3}$$

where $D_{\text{KL}}(\cdot \,||\, \cdot)$ is the KL divergence and $\widehat{y}_i$ is the prediction $f_i(x)$ for $x \sim \mathcal{D}_\text{T}$. In practice, we optimize this quantity using empirical estimates of the distributions $p(\widehat{y}_i, \widehat{y}_j)$ and $p(\widehat{y}_i) \otimes p(\widehat{y}_j)$.

To prevent functions from collapsing to degenerate solutions such as predicting a single label for the entire target set while maintaining good source accuracy, we also minimize an optional regularization loss which regularizes the marginal prediction of each head across the target dataset:

$$\mathcal{L}_{\text{reg}}(f_i) = D_{\text{KL}}\left(p(\widehat{y}_i) \,||\, p(y)\right), \tag{4}$$

where $p(y)$ is a hyperparameter. Our experiments in Fig. 14 show that DivDis is not very sensitive to the choice of this hyperparameter. If we expect the label distribution of the source and target datasets to be similar, we can simply set $p(y)$ to be the label distribution in the source dataset $\mathcal{D}_\text{S}$.

The overall objective for the DIVERSIFY stage is a weighted sum with hyperparameters $\lambda_1, \lambda_2 \in \mathbb{R}$:

$$\sum_i \mathcal{L}_{\text{xent}}(f_i) + \lambda_1 \sum_{i \neq j} \mathcal{L}_{\text{MI}}(f_i, f_j) + \lambda_2 \sum_i \mathcal{L}_{\text{reg}}(f_i). \tag{5}$$

We note that the quantities needed for the mutual information term (3) is easily computed in parallel across examples within a batch using modern deep learning libraries; we show a code snippet in Appendix B. In practice, the cost of computing the objective (5) is dominated by the cost of feeding two batches—one source and one target—to the network. The time- and space- complexity of one step in the DIVERSIFY stage is approximately $\times 2$ compared to a standard SGD step in optimizing ERM with the source data, and both can be reduced by using a smaller batch size.

## 3.3 DISAMBIGUATE : SELECT THE BEST HEAD

After learning a diverse set of functions that all achieve good training performance in the DIVERSIFY stage, we DISAMBIGUATE by selecting one of the functions. For example, once our model has learned both the animal and background classifiers for the cow-camel task, we can quickly see which is right by observing the ground-truth label of an image of a cow in the desert. As such an example is not present in the given labeled source or unlabeled target data, this stage requires information beyond what was used in the first stage. We now present three different strategies for head selection during the DISAMBIGUATE stage.

**Active querying.** To select the best head with a minimal amount of supervision, we propose an active querying procedure, in which the model acquires labels for the most informative subset of the unlabeled target dataset $\mathcal{D}_T$. Since larger difference in predictions indicates more information for disambiguation, we sort each target datapoint $x \in \mathcal{D}_T$ according to the total distance between head predictions $\sum_{i \neq j} |f_i(x) - f_j(x)|$. We select a small subset of the target dataset, which has the $m$ datapoints (i.e. $m \ll |\mathcal{D}_T|$) with the highest value of this metric. We measure the accuracy of each of the $N$ heads with respect to this labeled set and select the head with the highest accuracy.

**Random querying.** A related alternative to the active querying strategy is random querying, in which we label a random subset of the target dataset $\mathcal{D}_T$. Beyond its simplicity, an advantage of this procedure is that one can perform labeling in advance, because the datapoints to be labeled do not depend on the results of the DIVERSIFY stage. However, random querying is substantially less label-efficient than active querying because the set will likely include unambiguous datapoints for which labels are less informative for head selection.

**Disambiguation on source data.** Even if two functions are near-optimal in terms of labels, other properties of the source data $\mathcal{D}_S$ can distinguish between the two. Visualization techniques such as Grad-CAM (Selvaraju et al., 2017) reveal which region of the input is relevant to each head. Such regions can be compared against the true predictive features by a human observer. This comparison could also be performed in an automated way using existing bounding box annotations or pixel-wise segmentation labels, such as those available for the ImageNet and COCO datasets (Deng et al., 2009; Lin et al., 2014). A key advantage of this method is that we do not require anything from the target domain beyond unlabeled datapoints.

We note that the active and random query strategies do not use any information unavailable to previous approaches: existing methods (Nam et al., 2020; Liu et al., 2021) use labeled target data to tune hyperparameters. Unless stated otherwise, we use active querying because of its superior label efficiency. In Appendix E, we further analyze such label efficiency through a generalization bound that depends only on the number of heads and the difference between the best and second best head. In our experiments, the active strategy required as little as a single label. We emphasize that as long as the best head is selected, the choice of disambiguation method only affects label efficiency, and the final performance of the selected head is the same for all three methods. We summarize the overall structure of DivDis in Algorithm 1.

## 4 EXPERIMENTS

Through our experimental evaluation, we aim to answer the following questions. (1) What functions does the DIVERSIFY stage discover on simple, low-dimensional problems? (2) Can DivDis tackle image and language classification problems with severe underspecification, in which simplicity bias hinders the performance of existing approaches? (3) How sensitive is DivDis to hyperparameters, and what data assumptions are needed to tune them? (4) How does DivDis compare to unsupervised domain adaptation algorithms, which also leverage unlabeled data from the target domain?

Unless stated otherwise, DivDis uses a network with 2 heads and the DISAMBIGUATE stage uses the active querying strategy with 16 labels, which we found suffices to recover the best between two heads. We closely follow the experimental settings of previous works, and all experimental details including datasets, architectures, and hyperparameters are in the appendix.

### 4.1 ILLUSTRATIVE TOY TASK

**2D classification task.** We start with a synthetic 2D binary classification problem to see what functions the DIVERSIFY stage discovers. The task is shown in Fig. 2 (left): inputs are points on a plane, and the source distribution includes points in the second and fourth quadrants while the target distribution covers all four quadrants. By design, the labels for the points in the first and third quadrants are ambiguous. We train a network with two heads and measure target accuracy of each head throughout the DIVERSIFY stage. We visualize learning curves and decision boundaries in Fig. 3, which show that while the two heads initially learn to fit the source data, they later diverge to functions based on different predictive features. We also show an extended visualization with additional metrics in the appendix (Fig. 8).

**Coverage of near-optimal set.** To further understand how well DivDis can cover the set of near-optimal functions, we trained a 20-head model on the same 2D classification task, where each head

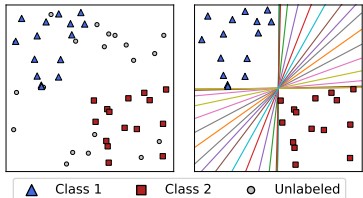

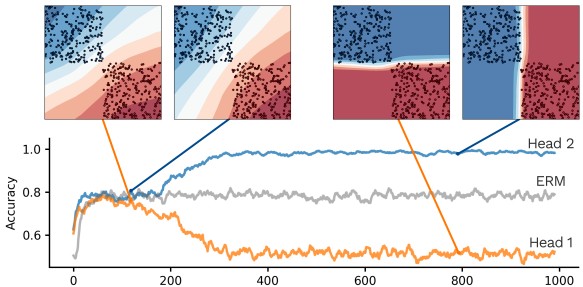

Figure 2: Left: synthetic 2D classification task with underspecification. Right: decision boundaries of 20 linear functions discovered by the DIVERSIFY stage. Together, these functions span the set of linear decision boundaries consistent with the data.

Figure 3: Learning curves for DivDis with 2 heads trained on the synthetic 2D classification task. We show the target domain accuracy of each head, along with that of ERM for comparison. The two heads initially represent similar functions, but later on diverge to represent classifiers that rely on entirely different features while remaining consistent with the source data. Head 2 achieves substantially higher accuracy than ERM.

is a linear classifier. Results in Fig. 2 (right) show that the heads together span the set of linear classifiers consistent with the source data. Note that this set includes the function with the diagonal decision boundary ($y = x$). This result suggests that given enough heads, the set of functions learned by DivDis can sufficiently cover the set of near-optimal functions, including the simplest function learned by existing methods.

**Comparison with ensembles.** We compare the diversity of the functions produced by the DIVERSIFY stage to that of independently trained models on a 3-dimensional version of the binary classification task. We measure how much each function relies on each of the three input dimensions through the Pearson correlation coefficient between each input dimension and the prediction. Visualizations in Fig. 9 of Appendix D show that the functions learned by DivDis depend on different input features, whereas independently trained models use a roughly equal mix of all features. This experiment demonstrates that the diversity in a vanilla ensemble cannot effectively cover the set of near-optimal functions, and is therefore insufficient for underspecified problems.

## 4.2 TASKS WITH COMPLETE SPURIOUS CORRELATION

We evaluate DivDis on datasets with a *complete correlation*, where the source distribution has a spuriously correlated attribute that can predict the label perfect accuracy. To make this problem tractable, we leverage unlabeled target data $\mathcal{D}_T$ for which the spurious attribute is not completely correlated with labels, as in the toy classification task (Fig. 2). Introducing complete correlations makes the problem considerably harder than existing subpopulation shift problems, because even the $\varepsilon$-optimal set with $\varepsilon = 0$ includes classifiers based on the spurious attribute.

**Real data with complete correlation.** We evaluate DivDis on several benchmarks for spurious correlation (Waterbirds, CelebA, MultiNLI) that we modified to exhibit a complete correlation between the label and the spurious attribute. Specifically, we alter the source dataset to include only the majority groups (e.g. waterbird with water background) and (e.g. landbird with land background). We use the original target validation set as the unlabeled data for the DIVERSIFY stage. We denote these tasks as Waterbirds-CC etc., to distinguish from the original benchmarks. These tasks are considerably more difficult than the original, and they introduce a specific challenge not addressed in the existing literature: leveraging the difference in source and target data distribution to encode and subsequently disambiguate tasks with high degrees of underspecification. To our best knowledge, no prior methods are designed to address such complete correlations.

To examine the sample efficiency of the additional label queries used by DivDis during the DISAMBIGUATE stage, we experiment with $N = \{4, 8, 16, 32, 64, 128\}$ labeled target examples on the Waterbirds-CC and CelebA-CC tasks, evaluating the performance using both active and random querying. We consider two baseline methods which similarly use $N$ datapoints from the target distribution in addition to the completely correlated source data: (1) ERM on the Waterbirds-CC training dataset with $N$ additional minority datapoints and (2) Deep Feature Reweighting (Kirichenko et al., 2022, DFR), which first trains an ERM model on the training dataset and then fine-tunes the last layer on a group-balanced set of size $N$. Fig. 4 shows that while DFR is substantially more sample-efficient than ERM, possibly due to its two-stage nature, even DivDis with random querying shows

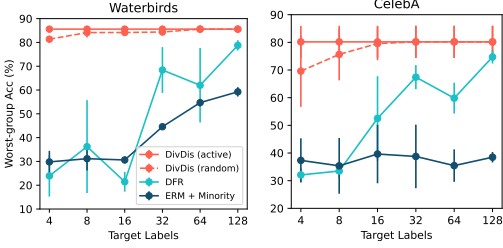
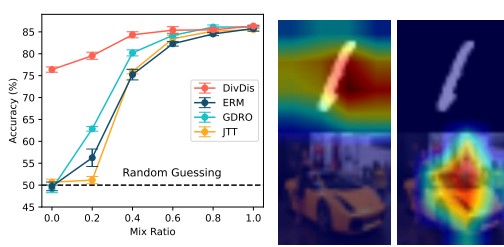

(a) Accuracy vs mix ratio.   (b) Grad-CAM visualization.

Figure 4: Worst-group accuracy on Waterbirds-CC and CelebA-CC data, given different numbers of labeled target examples. DivDis is significantly more label-efficient than other methods in learning from non-majority labels.

Figure 5: (a) Accuracy of DivDis, ERM, Group DRO, and JTT on MNIST-CIFAR data with different correlation ratios. (b) Grad-CAM visualization of two learned heads on a random datapoint from the MNIST-CIFAR source dataset. See Sec. 4.2 for details.

|  | Worst (%) |
|---|---|
| Ensemble | 34.8 ± 1.0 |
| Teney et al. (2021) | 33.4 ± 1.6 |
| Ensemble + Dis | 36.0 ± 1.2 |
| DivDis (ours) | **82.4 ± 1.9** |

Table 1: Worst-group accuracy of various ensemble methods on the Waterbirds-CC dataset. All methods use two heads or ensemble members.

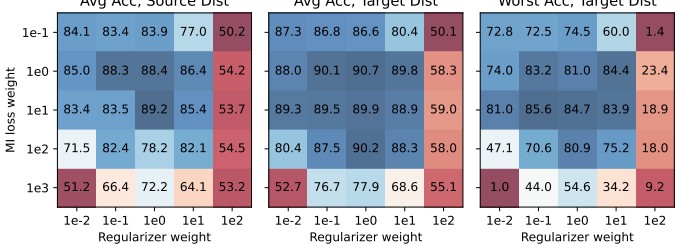

Figure 6: Hyperparameter grids for DivDis on the Waterbirds dataset. We show three metrics: average accuracy on the source and target distributions and worst-group accuracy on the target distribution. The high correlation between the three metrics indicates that we can tune the hyperparameters of DivDis using only held-out labeled source data.

substantially higher sample efficiency. This indicates that the small set of diverse functions learned during the DIVERSIFY stage is critical for quickly learning from additional minority data.

As an additional naïve point of comparison in the complete correlation setting, we evaluate the performance of existing methods for addressing spurious correlations: ERM, JTT (Liu et al., 2021), and Group DRO (Sagawa et al., 2020)). As these methods are designed for settings with a milder spurious correlation and do not leverage additional information through unlabeled target data, they are expected to fail in the complete correlation setting. Results for the Waterbirds-CC, CelebA-CC, and MultiNLI-CC tasks in Tab. 6 of Appendix D show that these methods show subpar performance: their worst-group accuracy is worse than that of random guessing in some settings, and have a performance gap with DivDis of up to 35% in worst-group accuracy. In Tab. 1, we also compare DivDis to a vanilla ensemble and our implementation of Teney et al. (2021), where we observe no further benefits from these ensemble methods in our complete correlation setting. This experiment also shows that DivDis achieves substantially higher worst-group accuracy compared to applying the Dis stage to an independent ensemble, demonstrating that that the Div stage is critical for performance.

**Overcoming simplicity bias on MNIST-CIFAR data.** In the MNIST-CIFAR task (Shah et al., 2020), each datapoint is a concatenation of one MNIST image and one CIFAR image, and labels are binary. The source dataset is completely correlated: the first class consists of (MNIST zero, CIFAR car) images, and the second class is (MNIST one, CIFAR truck). The unlabeled target dataset is constructed from the validation sets of MNIST and CIFAR, and has no such correlation. By design, the unseen combinations are ambiguous. We evaluate on variants of MNIST-CIFAR with different levels of underspecification. We denote the *mix ratio* of $\mathcal{D}_S$ as $r \in [0, 1]$, where $r = 0$ indicates completely correlated data as described above, and $r = 1$ indicates the distribution of the target set. Fig. 5a shows the target domain accuracy of DivDis, ERM, Group DRO (Sagawa et al., 2020), and JTT (Liu et al., 2021) after training with various mix ratios. Existing methods fail to do better than random guessing (50%) in the completely correlated setting (ratio= 0.0), whereas DivDis achieves over 75% accuracy. Higher ratios make the problem closer to an i.i.d. setting where the source and target distributions are identical, and the two methods achieve similar accuracy.

**Comparison of DISAMBIGUATE strategies on MNIST-CIFAR data.** Using MNIST-CIFAR data with a complete correlation, we plot the average final accuracy after the DISAMBIGUATE

| Tuned with: | Waterbirds Worst Acc | | CelebA Worst Acc | |
|---|---|---|---|---|
| | Worst | Average | Worst | Average |
| CVaR DRO | 75.9% | 62.0% | 64.4% | 36.1% |
| LfF | 78.0% | 44.1% | 77.2% | 24.4% |
| JTT | 86.7% | 62.5% | 81.1% | 40.6% |
| DivDis (ours) | 85.6% | **81.0%** | 55.0% | **55.0%** |

Table 2: Worst-group test accuracies in the Waterbirds and CelebA tasks, when tuning hyperparameters with respect to average and worst-group accuracies. DivDis is substantially more robust to hyperparameter choice in both tasks, allowing us to tune hyperparameters without group labels.

| | Test Acc |
|---|---|
| Pseudo-Label | $67.7 \pm 8.2$ |
| DANN | $68.4 \pm 9.2$ |
| FixMatch | $71.0 \pm 4.9$ |
| CORAL | $77.9 \pm 6.6$ |
| NoisyStudent | $86.7 \pm 1.7$ |
| DivDis (ours) | $\mathbf{90.4} \pm 1.8$ |

Table 3: OOD test accuracy on Camelyon17-WILDS. All methods leverage unlabeled target data.

stage for both the active query and random query strategies, for different number of labels used. Fig. 11 of Appendix D shows that active querying in particular is very efficient, and one label suffices for finding the head with highest target data accuracy. We further verify the possibility of disambiguation on source data: Fig. 5b shows Grad-CAM (Selvaraju et al., 2017) visualizations of two heads on a randomly sampled image from the source dataset. Even though the two heads predict the same label, they respectively focus on distinct features of the data: the MNIST region and the CIFAR region. Since we know that the true predictive feature is the CIFAR image, we can select the second head based on this single datapoint, and nothing beyond what was used during training.

### 4.3 Underspecification from Distribution Shift

**Comparison with unsupervised domain adaptation methods.** Finally, we evaluate DivDis on the Camelyon17-WILDS dataset (Sagawa et al., 2022), a large-scale tumor classification dataset consisting of $455,954$ labeled source datapoints and $600,030$ unlabeled target datapoints. The target datapoints are collected from a hospital not seen in the source data. As the unlabeled set for DivDis, we use the official val_unlabeled set provided by Sagawa et al. (2022). We compare against several approaches that can also leverage this unlabeled data: Pseudo-Label (Lee et al., 2013), FixMatch (Sohn et al., 2020), CORAL (Sun et al., 2016), and NoisyStudent (Xie et al., 2019). All results other than DivDis are from Sagawa et al. (2022). Quantitative results in Tab. 3 show that DivDis outperforms these methods, achieving over $90\%$ OOD test set accuracy. This experiment demonstrates that the approach of DivDis scales to large datasets, effectively capturing the implicit ambiguity inside large unlabeled datasets from a related but different distribution. This experiment demonstrates that DivDis can effectively leverage unlabeled data for underspecification arising from variation in real-world data collection conditions.

**Do we need group labels for hyperparameter tuning?** Existing methods for learning from data with subpopulation shift typically tune hyperparameters using group label annotations (Levy et al., 2020; Nam et al., 2020; Liu et al., 2021), making them deployable only in scenarios where group labels are available. To examine the dataset assumptions required to successfully tune DivDis's hyperparameters, we ran a hyperparameter sweep over $(\lambda_1, \lambda_2)$. For each setting, we measure three metrics using held-out data: (1) average accuracy on $\mathcal{D}_S$, (2) average accuracy on $\mathcal{D}_T$, and (3) worst-group accuracy on $\mathcal{D}_T$. These metrics correspond to different assumptions about available information: (1) labeled source data, (2) labeled target data, and (3) labeled target data with group annotations, respectively.

Results in Fig. 6 show that the three metrics have a clear correlation. Notably, this implies that that tuning the hyperparameters of DivDis with respect to average accuracy on $\mathcal{D}_S$ yields close an optimal model for worst-group accuracy on the target distribution. Additional experiments on the CelebA dataset exhibit a similar trend, as shown in Appendix D. In Tab. 2, we compare DivDis to existing methods as reported by Liu et al. (2021). Note that the "Average" columns here correspond to our second weakest data assumption of labeled target data, whereas Fig. 6 implies that we can tune DivDis's hyperparameters using only labeled source data. This experiment demonstrates that compared to previous methods for distribution shift, DivDis's hyperparameters require substantially less information to tune.

## 5 Related Work

**Underspecification.** Prior works have discussed the underspecified nature of many datasets (D'Amour et al., 2020; Oakden-Rayner et al., 2020). Underspecification is especially problem-

atic when the bias of deep neural networks towards simple functions (Arpit et al., 2017; Gunasekar et al., 2018; Shah et al., 2020; Geirhos et al., 2020; Pezeshki et al., 2021;?) is not aligned with the true function. Yet, these works do not present a general solution. As we find in Section 4, DivDis can address underspecified datasets, even when one viable solution is much simpler than another, since only one of the heads can represent the simplest solution. Our notion of near-optimal sets can be seen as an extension of *Rashomon sets* (Fisher et al., 2019; Semenova et al., 2019) to the unsupervised domain adaptation setting. Active learning methods (Cohn et al., 1996; Hanneke et al., 2014) are also related in that they handle underspecification by reducing ambiguity. Our MI-based diversity term resembles a common active learning criterion (Houlsby et al., 2011), but a key difference is that we directly optimize a set of models with respect to our criterion.

**Ensemble methods.** Our approach is related to ensemble methods (Hansen & Salamon, 1990; Dietterich, 2000; Lakshminarayanan et al., 2017), which aggregate the predictions of multiple learners. Ensembles have been shown to perform best when each member produces errors independently of one another (Krogh et al., 1995), a property we exploit by maximizing disagreement on unlabeled test data. Previous works have extended ensembles by learning a diversified set of functions (Pang et al., 2019; Parker-Holder et al., 2020; Wortsman et al., 2021; Rame & Cord, 2021; Sinha et al., 2021). While the DIVERSIFY stage similarly learns a collection of diverse functions, our approach differs in that we directly optimize for diversity on a separate target dataset. Prior works have learned a diverse set of reinforcement learning policies both with an underlying task Mouret & Clune (2015); Conti et al. (2018); Kumar et al. (2020) and with no task at all Lehman & Stanley (2011); Eysenbach et al. (2019); Sharma et al. (2020). While similarly motivated, these works operate in a very different setting, since we consider supervised learning with underspecification. So-called *quality diversity* methods aim to balance performance and novelty in a population Lehman & Stanley (2011); Mouret & Clune (2015); Cully & Demiris (2017).

Two recent works leverage unlabeled target data to learn a set of diverse functions. Teney et al. (2021) introduces a gradient orthogonality constraint with respect to features from a pre-trained backbone. We provide further experimental comparison to this method in Appendix A. Concurrently to our work, Pagliardini et al. (2022) propose to sequentially train a set of functions with a diversity loss on target data. In contrast, DivDis requires a single network and training loop regardless of the number of heads. Furthermore, Sec. 4 demonstrates that DivDis scales to larger datasets.

**Robustness and causality.** Many recent methods aim to produce robust models that succeed even in conditions of distribution shift (Tzeng et al., 2014; Ganin et al., 2016; Arjovsky et al., 2019; Sagawa et al., 2020; Nam et al., 2020; Creager et al., 2021; Liu et al., 2021). While our work is similarly motivated, we address a class of problems that these previous methods fundamentally cannot handle. By nature of learning only one function, these robustness methods cannot disambiguate problems where the true function is truly ambiguous, in the sense that functions based on two different features can both be near-optimal. DivDis handles such scenarios by learning multiple functions in the DIVERSIFY stage and then choosing the correct one in the DISAMBIGUATE stage with minimal added supervision. This research direction is also related to inferring the causal structure (Pearl, 2000; Schölkopf, 2019) of observed attributes. Although many causality works focus on situations in which interventions are impossible, we explore inherently ambiguous problems where some form of intervention is necessary to succeed. Additionally, recent methods for extracting causality from observational data have been most successful in low-dimensional settings (Louizos et al., 2017; Goudet et al., 2018; Ke et al., 2019), whereas our method easily scales to large convolutional networks for image classification problems.

## 6 CONCLUSION

We proposed Diversify and Disambiguate (DivDis), a two-stage framework for learning from underspecified data. Our experiments show that DivDis has substantially higher performance when learning from datasets with high degrees of underspecification, at the modest cost of unlabeled target data and a few corresponding labels. To our knowledge, our method is the first to address this problem setting in the context of underspecification. An appealing property of DivDis is its automatic discovery of meaningful complementary features, as demonstrated in Fig. 6 and Tab. 2. This capability is related to that of disentangled feature learning (Chen et al., 2016; Higgins et al., 2017; Kim & Mnih, 2018; Chen et al., 2018), which is often posed as an unsupervised problem. Our experiments show the alternative possibility of learning different possible meanings of labels in an underspecified supervised learning setting.

## ACKNOWLEDGEMENTS

We thank Damien Teney for help in reproducing the published Collages task experiments. We also thank Pang Wei Koh, Henrik Marklund, Annie S. Chen, other members of the IRIS and RAIL labs, and anonymous reviewers for helpful discussions and feedback. This work was supported in part by KFAS, Google, Apple, Juniper Networks, and Open Philanthropy. Chelsea Finn is a fellow in the CIFAR Learning in Machines and Brains program.

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

| Method | Parameters | MNIST | SVHN | FMNIST | CIFAR |
|---|---|---|---|---|---|
| Upper bound oracle | - | 99.7 | 89.7 | 77.4 | 68.7 |
| Evading (N=8) | 27152 | 97.3 | 82.1 | 59.6 | 55.8 |
| Evading (N=16) | 54304 | 96.6 | 72.1 | 64.6 | **58.4** |
| Evading (N=16, weight sharing) | 3904 | 99.7 | 50.8 | 50.3 | 50.2 |
| Evading (N=32, weight sharing) | 4448 | 99.6 | 50.7 | 50.1 | 50.2 |
| DivDis (N=2) | 3428 | 98.1 | 66.5 | 62.5 | 51.5 |
| DivDis (N=4) | 3496 | 99.4 | **75.1** | **66.1** | 53.4 |

Table 4: Accuracies for the 4-way collage task. We denote the method in Teney et al. (2021) as "Evading". DivDis with 4 heads shows competitive performance to the Evading method with up to 16 independent models. We note that by default, Evading uses *separate models*, meaning parameter count and computation scales linearly with $N$. The Evading method fails when the $N$ networks share backbone parameters.

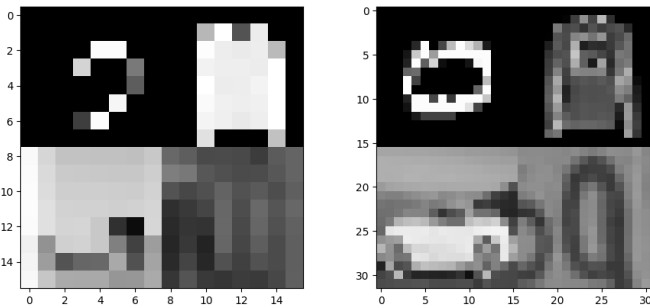

Figure 7: Samples from the original $16 \times 16$ collage dataset from Teney et al. (2021) (left) and the higher-resolution $32 \times 32$ dataset used in Tab. 5 (right). A substantial amount of visual detail is lost in the original dataset, especially in harder datasets such as FashionMNIST and CIFAR.

## A  THE COLLAGES TASK: CAN DIVDIS LEARN MORE THAN TWO FUNCTIONS?

To further investigate whether DivDis can learn more than two functions, we evaluate DivDis on the four-block collages dataset (Teney et al., 2021). This dataset is a binary classification task in which four attributes have a complete spurious correlation with labels in the source dataset. Each datapoint is constructed by concatenating one image each from the MNIST (Deng, 2012), SVHN (Netzer

| Method | MNIST | SVHN | FMNIST | CIFAR |
|---|---|---|---|---|
| Evading (N=8) | 97.3 | 82.1 | 59.6 | 55.8 |
| Evading (N=16) | 96.6 | 72.1 | 64.6 | 58.4 |
| Evading (N=8, ours) | 99.5 | 70.0 | 56.3 | 50.9 |
| Evading (N=16, ours) | 99.7 | 86.3 | 65.9 | 60.8 |
| Evading (N=8, ours, ×2) | 99.5 | 49.7 | 51.6 | 49.7 |
| Evading (N=16, ours, ×2) | 99.8 | 49.6 | 52.0 | 49.7 |
| DivDis (N=2, ×2) | 97.6 | 63.9 | 61.8 | 51.7 |
| DivDis (N=4, ×2) | 96.5 | **73.0** | **69.2** | **52.7** |

Table 5: Collage dataset accuracies with higher resolution images ($16 \times 16 \rightarrow 32 \times 32$). We re-implemented the "Evading" method of Teney et al. (2021), and denote results from our codebase as "ours". The Evading method fails on $32 \times 32$ images, likely because input-space diversification is sensitive to input dimensionality. In contrast, DivDis performs output-space diversification and is robust to change in input dimensionality.

et al., 2011), FashionMNIST (Xiao et al., 2017), and CIFAR (Krizhevsky et al., 2009) datasets in a $2 \times 2$ grid. This task aims to learn the predictive patterns corresponding to each of the four datasets based on labeled data with a complete correlation between all four. After training on the source data, we separately test for each of the four features using de-correlated datasets labeled according to one of the four features. We report the highest accuracy achieved among functions inside the learned ensemble for each of the four features.

We evaluated DivDis with 2 or 4 heads on this task. For more than two heads, we minimize the average mutual information loss (3) between all pairs of heads. As the unlabeled set for DivDis, we use decorrelated 4-way collage data constructed from the validation split of each dataset. Results in Tab. 4 compare feature-wise test accuracies with that of "Evading" (Teney et al., 2021). DivDis with 4 heads shows competitive performance to Evading with up to 16 independent models, indicating that DivDis requires substantially fewer functions to achieve the same performance as Evading. We also note that it is crucial to use the correct number of heads for DivDis. As expected, training only $N = 2$ heads results in subpar performance because this dataset has 4 different features. The two methods differ in scalability: Evading uses $N$ independent models, meaning that the number of parameters and computation cost scales linearly with $N$. In contrast, DivDis shares all weights except for the last layer. As shown in Tab. 4, Evading fails when the $N$ networks perform weight sharing.

As shown in Fig. 7, the images in the original collages dataset have resolution $16 \times 16$, which loses a substantial amount of visual detail because each image from (MNIST, SVHN, FMNIST, CIFAR) is reduced to an $8 \times 8$ image. We evaluate both methods on larger $32 \times 32$ collage data in Tab. 5, where Evading fails to discover features besides MNIST. Evading is an *input-space diversification* method, making it sensitive to input dimensionality. In contrast, DivDis performs *output-space diversification*, making its performance more robust to changes in input dimensionality.

Diversification is an effective approach to handling underspecification, and the two broad approaches of input-space diversification (Teney et al., 2021; 2022) and output-space diversification (e.g., DivDis) are complementary approaches with different strengths and weaknesses. As shown in Tables 4 and 5, DivDis is more scalable in terms of model count and input size. We also note that this scalability is also reflected in results on the large-scale Camelyon dataset: DivDis achieves $90.4\%$ accuracy on test hospitals (Tab. 3), whereas the method in Teney et al. (2022) achieves $82.5\%$. Conversely, while DivDis was shown to require fewer functions, Evading (Teney et al., 2021) achieves higher performance on the original collage dataset when using up to 96 models. An exciting direction for future work is understanding and consolidating these tradeoffs between input-space and output-space diversification methods for underspecified tasks.

## B  PARALLEL IMPLEMENTATION OF MUTUAL INFORMATION OBJECTIVE

```python
import torch
from einops import rearrange

def mutual_info_loss(probs):
    """ Input: predicted probabilites on target batch. """
    B, H, D = probs.shape  # B=batch_size, H=heads, D=pred_dim
    marginal_p = probs.mean(dim=0)  # H, D
    marginal_p = torch.einsum("hd,ge->hgde", marginal_p, marginal_p)  # H, H, D, D
    marginal_p = rearrange(marginal_p, "h g d e -> (h g) (d e)")  # H^2, D^2

    joint_p = torch.einsum("bhd,bge->bhgde", probs, probs).mean(dim=0)  # H, H, D, D
    joint_p = rearrange(joint_p, "h g d e -> (h g) (d e)")  # H^2, D^2

    kl_divs = joint_p * (joint_p.log() - marginal_p.log())
    kl_grid = rearrange(kl_divs.sum(dim=-1), "(h g) -> h g", h=H)  # H, H
    pairwise_mis = torch.triu(kl_grid, diagonal=1) # Get only off-diagonal KL divergences
    return pairwise_mis.mean()
```

This implementation is based on the PyTorch Paszke et al. (2019) and einops Rogozhnikov (2022) libraries. It demonstrates that the mutual information objective (3) is easily parallelized across the input batch using standard tensor operations.

## C    EXPERIMENTAL SETUP

### C.1    DETAILED DATASET DESCRIPTIONS

**Toy classification task.** Our toy binary classification data is constructed as follows. The source dataset $\mathcal{D}_S$ has binary labels with equal aggregate probability $p(y = 0) = p(y = 1) = \frac{1}{2}$. Each datapoint is a 2-dimensional vector, and the data distribution for each class in the source dataset is:

$$p(x \mid y = 0) = \text{Unif}([-1, 0] \times [0, 1])$$
$$p(x \mid y = 1) = \text{Unif}([0, 1] \times [-1, 0]).$$

In contrast, the data distribution for each class in the target dataset is:

$$p(x \mid y = 0) = \text{Unif}([-1, 0] \times [-1, 1])$$
$$p(x \mid y = 1) = \text{Unif}([0, 1] \times [-1, 1]).$$

Labels are balanced for the target dataset. Put differently, the target dataset has a larger span than the source dataset, and the labels of the target dataset reveal that the true decision boundary is the $Y$-axis.

**CXR-14 pneumothorax classification.** The CXR-14 dataset Wang et al. (2017) is a large-scale dataset for pathology detection in chest radiographs. We evaluate on the binary pneumothorax classification task, which has been reported to suffer from hidden stratification: a subset of the images with the disease include a chest drain, a common treatment for the condition Oakden-Rayner et al. (2020).

**Waterbirds dataset.** Each image in the Waterbirds dataset is constructed by pasting a waterbird or landbird image to a background drawn from the Places dataset Zhou et al. (2017). There are two backgrounds in this dataset – water and land, where each category of birds is spuriously correlated with one background. Specifically, there are 4,795 training samples, where 3,498 samples are from "waterbirds in water" and 1,057 samples are from "landbirds in land". "Waterbirds in land" and "landbirds in water" are considered as minority groups, where 184 and 56 samples are included, respectively.

**CelebA dataset.** The CelebA dataset Liu et al. (2015) is a large-scale image dataset with over $200,000$ images of celebrities, each with 40 attribute annotations. We construct four different completely correlated problem settings, each based on a pair of attributes. The pair of attributes consists of a label attribute and a spurious attribute, and we remove all examples from the two minority groups in the source dataset. The four problem settings are summarized below. Our task construction is similar to that of Sagawa et al. (2020), which uses hair color as the label and gender as the spurious attribute.

|            | Label attribute     | Spurious attribute |
|------------|---------------------|--------------------|
| CelebA-CC-1 | Mouth_Slightly_Open | Wearing_Lipstick   |
| CelebA-CC-2 | Attractive          | Smiling            |
| CelebA-CC-3 | Wavy_Hair           | High_Cheekbones    |
| CelebA-CC-4 | Heavy_Makeup        | Big_Lips           |

**MultiNLI dataset.** Given a hypothesis and a promise, the task of MultiNLI dataset is to predict if the hypothesis is entailed by, neutral with, or contradicts with the promise. The spurious correlation exists between contradictions and the presence of the negation words nobody, no, never, and nothing Gururangan et al. (2018). The whole MultiNLI dataset is divided into six groups, where each spurious attribute belongs to {"no negation", "negation"} and each label belongs to {entailed, neutral, contradictory}. There are 206,175 samples in total, where the smallest group only has 1,521 samples (entailment with negations).

**Camelyon17-WILDS dataset.** This dataset is part of the U-WILDS benchmark Sagawa et al. (2022). Input images of patches from lymph node sections are given, and the task is to classify as either a tumor or normal tissue. Evaluation is performed on OOD hospitals for which labels are unseen during training. The model is given unlabeled validation images from the OOD hospitals.

## C.2 DivDis Hyperparameter Settings

We show below the hyperparameters used in our experiments:

|  | Toy Clasification | MNIST-CIFAR | Waterbirds | CXR-14 | Waterbirds-CC | CelebA-CC | MultiNLI-CC |
|---|---|---|---|---|---|---|---|
| $N$ | $2, 20$ | $2$ | $2$ | $2$ | $2$ | $2$ | $2$ |
| $\lambda_1$ | $10$ | $10$ | $1, 10, 100, 1000$ | $10$ | $10$ | $10$ | $1000$ |
| $\lambda_2$ | $10$ | $10$ | $0.1, 1, 10, 100$ | $10$ | $0, 10$ | $0, 10$ | $0, 0.1$ |
| $m$ | $1$ | $1$ | $16$ | $16$ | $16$ | $16$ | $16$ |

## D Additional Experiments

**Extended learning curves on toy task.** In Fig. 8, we show an extended version of the learning curve shown in Fig. 3. The extended plot includes cross-entropy loss and mutual information loss during training. The learning curves show that cross-entropy loss decreases first, at which point both of the heads represent functions similar to the ERM solution. Afterwards, the mutual information loss decreases, causing the functions represented by the two heads to diverge.

**Visualization of functions on 3D toy task.** We examine the extent to which the DIVERSIFY stage can produce different functions by visualizing which input dimension each function relies on. We modify the synthetic binary classification task to have 3-dimensional inputs and train DivDis with $\{2, 3, 5\}$ heads. For each head, we visualize the Pearson correlation coefficient between each input dimension and output. We normalize this 3-dimensional vector to sum to one and plot each model as a point on a 2-simplex in Fig. 9, with independently trained functions as a baseline. The results show that the DIVERSIFY stage acts as a repulsive force between the functions in function space, allowing the collection of heads to explore much closer to the vertices. This experiment also demonstrates why vanilla ensembling is insufficient for underspecified problems: the diversity due to random seed is not large enough to effectively cover the set of near-optimal functions.

**Noisy dimension.** To see if DivDis can effectively combat simplicity bias, we further evaluate on a harder variant of the 2D classification problem in which we add noise along the x-axis. This noise makes the "correct" decision boundary have positive non-zero risk, making it harder to learn than the other function. Results in Fig. 10 demonstrate that even in such a scenario, DivDis recovers both the x-axis and y-axis decision boundaries, suggesting that DivDis can be effective even in scenarios where ERM relies on spurious features due to simplicity bias.

**Evaluation on completely correlated data.** We evaluate DivDis on existing benchmarks modified to exhibit complete correlation. Using the Waterbirds Sagawa et al. (2020), CelebA Liu et al. (2015), and MultiNLI Gururangan et al. (2018) datasets, we alter the source dataset to include only majority groups while keeping target data intact. We denote these tasks as Waterbirds-CC, MultiNLI-CC, etc to distinguish from the original benchmarks. These -CC tasks are considerably more difficult than the original benchmarks, and introduce a specific challenge not addressed in the existing literature: leveraging the difference in source and target data distribution to encode and subsequently disambiguate tasks with high degrees of underspecification. To our best knowledge, no prior methods are designed to address such complete correlations.

As the closest existing problem setting is subpopulation shift, we show the performance of ERM, JTT Liu et al. (2021), and Group DRO Sagawa et al. (2020) as a naive point of comparison. We also include a random guessing baseline as a lower bound on performance. As expected, Tab. 6 shows that existing subpopulation shift methods show subpar performance on these tasks, notably failing to do better than random guessing in the Waterbirds-CC task. This is hardly surprising, as methods based on loss upweighting such as JTT and Group DRO require minority examples in the training data to upweight. In contrast, DivDis is well-suited to this challenging setting, and can deal with complete correlation by leveraging unlabeled target data to find different predictive features of the labels.

**CelebA hyperparameter grid.** In Fig. 13, we show an additional hyperparameter grid for the CelebA dataset. This grid shows a strong corrleation between metrics with respect to hyperparameter choice, indicating that DivDis can be tuned using only labeled source data.

|  | Waterbirds-CC | | CelebA-CC-1 | | CelebA-CC-2 | | MultiNLI-CC | |
|---|---|---|---|---|---|---|---|---|
|  | Avg (%) | Worst (%) | Avg (%) | Worst (%) | Avg (%) | Worst (%) | Avg (%) | Worst (%) |
| Random | 50.0 | 50.0 | 50.0 | 50.0 | 50.0 | 50.0 | 33.3 | 33.3 |
| ERM | $60.5 \pm 1.6$ | $7.0 \pm 1.5$ | $70.9 \pm 2.0$ | $57.0 \pm 5.8$ | $73.1 \pm 0.9$ | $41.1 \pm 2.6$ | $53.2 \pm 1.5$ | $22.8 \pm 2.5$ |
| JTT | $44.6 \pm 1.9$ | $26.5 \pm 1.4$ | $71.4 \pm 1.9$ | $51.2 \pm 5.4$ | $78.7 \pm 0.8$ | $59.8 \pm 1.1$ | $80.0 \pm 4.0$ | $40.5 \pm 2.3$ |
| Group DRO | $55.6 \pm 4.8$ | $47.1 \pm 8.9$ | $71.6 \pm 0.3$ | $59.3 \pm 2.6$ | $71.6 \pm 2.4$ | $61.3 \pm 2.3$ | $79.1 \pm 3.4$ | $39.8 \pm 1.4$ |
| DivDis - reg | $87.2 \pm 0.8$ | $77.5 \pm 4.7$ | $91.0 \pm 0.4$ | $\mathbf{85.9 \pm 1.0}$ | $79.7 \pm 0.4$ | $\mathbf{69.3 \pm 1.9}$ | $80.3 \pm 0.6$ | $67.6 \pm 4.0$ |
| DivDis | $87.6 \pm 1.4$ | $\mathbf{82.4 \pm 1.9}$ | $90.8 \pm 0.4$ | $85.6 \pm 1.1$ | $79.5 \pm 0.2$ | $68.5 \pm 1.7$ | $79.9 \pm 1.2$ | $\mathbf{71.5 \pm 2.5}$ |

Table 6: Modified Waterbirds, CelebA, and MultiNLI datasets with complete correlation between labels and a spurious attribute. DivDis outperforms previous methods in terms of both average and worst-group accuracy.

|  | Acc | AUC | AUC (drain) | AUC (no-drain) |
|---|---|---|---|---|
| ERM | 0.883 | 0.828 | 0.904 | 0.717 |
| Pseudo-label | 0.898 | 0.835 | 0.904 | 0.721 |
| DivDis | 0.934 | 0.836 | 0.902 | **0.737** |

Table 7: Pneumothorax classification metrics on the test set of CXR-14. In addition to overall accuracy and AUC, we measure AUC separately on the two subsets of the positive class, drain and no-drain. DivDis shows higher AUC on the no-drain subset, which is more indicative of the intended population of patients not yet being treated. Performance gains on the no-drain subset also contribute positively to the overall metrics (Acc and AUC).

**Additional Disambiguate experiments on MNIST-CIFAR data.** In Fig. 11, we plot the accuracy of the head selected by the active and random querying strategies, when using different numbers of label queries. Notably, the active querying strategy successfully chooses the best head with even one label query. In Fig. 12, we show additional Grad-CAM plots for MNIST-CIFAR data, on 6 more random datapoints from the source dataset. Compared to the example given in the main text, these examples are just as informative in terms of which head is better.

**Effect of ratio.** We test various values between $0$ and $1$ for the regularizer loss (4), on the Waterbirds benchmark. Fig. 14 shows that even using a ratio of $0.1$ yields close to $80\%$ worst-group accuracy, demonstrating that the performance of DivDis is not very sensitive to this hyperparameter.

**CXR pneumothorax classification.** To investigate whether DivDis can disambiguate naturally occurring spurious correlations, we consider the CXR-14 dataset Wang et al. (2017), a large-scale dataset for pathology detection in chest radiographs. We evaluate on the binary pneumothorax classification task, which has been reported to suffer from hidden stratification: a subset of the images with the disease include a chest drain, a common treatment for the condition Oakden-Rayner et al. (2020). We train DivDis with two heads to see whether it can disambiguate between the visual features of chest drains and lungs as a predictor for pneumothorax. The unlabeled data for DivDis is a subset of validation data sampled so that the ratio of drain to no drain is 1:1. In addition to ERM, we compare against the semi-supervised learning method Pseudo-label Lee et al. (2013), to see how much of the performace gain of DivDis can be attributed to the unlabeled target set alone. In Tab. 7, we show test split accuracy and AUC, along with AUC for the subset of positive samples with and without a chest drain. Our experiments show that DivDis achieves higher AUC in the no-drain split while doing marginally worse on the drain split, indicating that the chosen head is relying more on the visual features of the lung. The overall metrics (Acc and AUC) indicate that this performance gain in the no-drain subset also leads to better performance in overall metrics.

## E  FINITE-HYPOTHESIS GENERALIZATION BOUND FOR HEAD SELECTION

**Proposition 1.** *Let the $N$ heads have risk $l_1 \leq l_2 \ldots \leq l_N \in \mathbb{R}$ on the target dataset, and let $\Delta = l_2 - l_1$. The required number of i.i.d. labels from the target set to select the best head with probability $\geq 1 - \delta$ is $m = \frac{2(\log 2N - \log \delta)}{\Delta^2}$.*

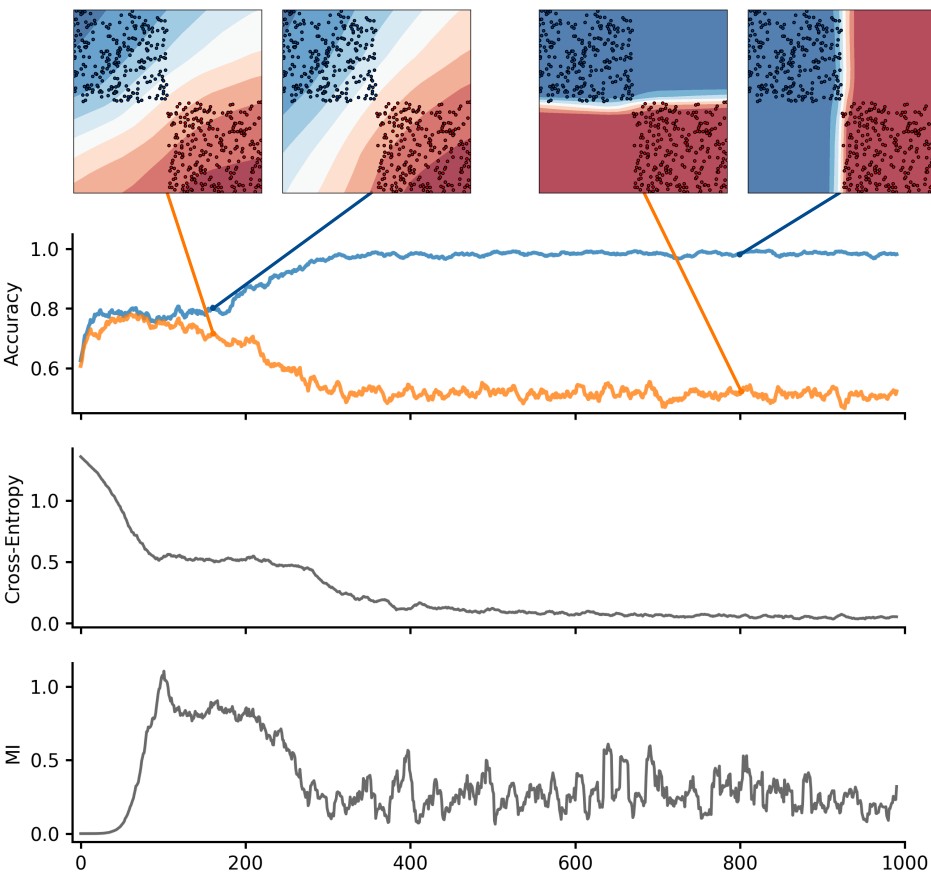

Figure 8: Extended visualization for the 2D classification task (Fig. 2), with additional curves for the cross-entropy and mutual information losses. Note that only accuracy is measured with target data, and the cross-entropy and mutual information losses are the training metrics for DIVERSIFY measured on source data. Until around iteration 100, the model initially decreases cross-entropy at the cost of increasing mutual information. The decision boundaries at this stage are similar for the two heads. Afterwards, both the mutual information and cross-entropy decrease, leading to the heads having very different decision boundaries.

Table 8: CelebA dataset with complete correlation between 4 different pairs of attributes. DivDis outperforms previous methods in all but one setting.

|  | CelebA-CC-1 | | CelebA-CC-2 | | CelebA-CC-3 | | CelebA-CC-4 | |
|---|---|---|---|---|---|---|---|---|
|  | Avg (%) | Worst (%) | Avg (%) | Worst (%) | Avg (%) | Worst (%) | Avg (%) | Worst (%) |
| ERM | $70.9 \pm 2.0$ | $57.0 \pm 5.8$ | $73.1 \pm 0.9$ | $41.1 \pm 2.6$ | $87.0 \pm 0.7$ | $71.9 \pm 2.6$ | $63.9 \pm 3.5$ | $23.0 \pm 1.4$ |
| JTT | $44.6 \pm 1.9$ | $26.5 \pm 1.4$ | $71.4 \pm 1.9$ | $51.2 \pm 5.4$ | $64.8 \pm 4.4$ | $34.0 \pm 10.2$ | $67.4 \pm 1.4$ | $49.3 \pm 8.2$ |
| GDRO | $71.6 \pm 0.3$ | $59.3 \pm 2.6$ | $71.6 \pm 2.4$ | $61.3 \pm 2.3$ | $88.2 \pm 0.6$ | $83.7 \pm 0.8$ | $65.0 \pm 1.6$ | $21.7 \pm 1.5$ |
| DivDis w/o reg | $91.0 \pm 0.4$ | $85.9 \pm 1.0$ | $79.7 \pm 0.4$ | $69.3 \pm 1.9$ | $79.5 \pm 0.6$ | $62.0 \pm 2.6$ | $84.7 \pm 0.5$ | $67.4 \pm 1.8$ |
| DivDis | $90.8 \pm 0.4$ | $85.6 \pm 1.1$ | $79.5 \pm 0.2$ | $68.5 \pm 1.7$ | $80.6 \pm 0.4$ | $67.1 \pm 1.9$ | $84.8 \pm 0.4$ | $73.5 \pm 2.6$ |

*Proof.* Given $m$ i.i.d. samples, Hoeffding's inequality gives us for all $\epsilon > 0$,

$$\mathbb{P}\left[ l - \hat{l} > \epsilon \right] \leq 2 \exp\left( -2m\epsilon^2 \right). \tag{6}$$

The event of failing to select the best head is a superset of the following event, for which we can bound the probability as:

$$\mathbb{P}\left[ \left( \left| l_1 - \widehat{l_1} \right| > \frac{\Delta}{2} \right) \vee \left( \left| l_2 - \widehat{l_2} \right| > \frac{\Delta}{2} \right) \vee \ldots \vee \left( \left| l_N - \widehat{l_N} \right| > \frac{\Delta}{2} \right) \right] \leq 2N \exp\left( -\frac{m\Delta^2}{2} \right). \tag{7}$$

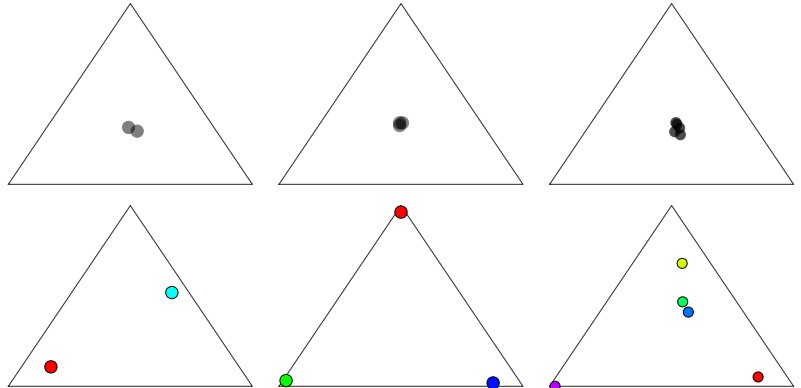

Figure 9: Visualization of $\{2, 3, 5\}$ functions trained independently (top row) and with DivDis (bottom row). Vertices of the 2-simplex represent the three dimensions of the input data. The functions learned by DivDis are much more diverse compared to independent training.

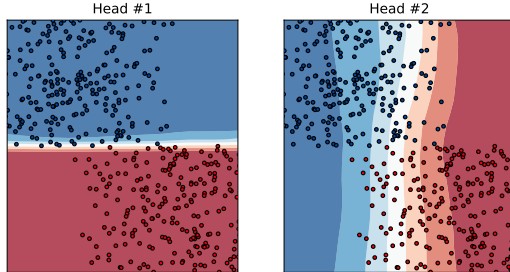

Figure 10: Functions learned by DivDis on a variant of the synthetic classification task, where the labeled source dataset has noise along the $x$-axis. The second head recovers the $Y$-axis decision boundary even though it is harder to learn due to the noise. This indicates that DivDis can successfully overcome simplicity bias and learn functions that ERM would not consider.

Solving for $\delta = 2N \exp\left(-\frac{m\Delta^2}{2}\right)$, we get the sample size bound

$$m^* = \frac{2(\log 2N - \log \delta)}{\Delta^2}. \tag{8}$$

$\square$

Table 9: CXR dataset test set metrics

|             | Accuracy          | AUC               | AUC (drain)       | AUC (no-drain)    |
| ----------- | ----------------- | ----------------- | ----------------- | ----------------- |
| ERM         | $0.883 \pm 0.006$ | $0.828 \pm 0.001$ | $0.904 \pm 0.008$ | $0.717 \pm 0.005$ |
| Pseudolabel | $0.898 \pm 0.015$ | $0.835 \pm 0.004$ | $0.904 \pm 0.007$ | $0.721 \pm 0.007$ |
| DivDis      | $0.934 \pm 0.014$ | $0.836 \pm 0.007$ | $0.902 \pm 0.006$ | $0.737 \pm 0.001$ |

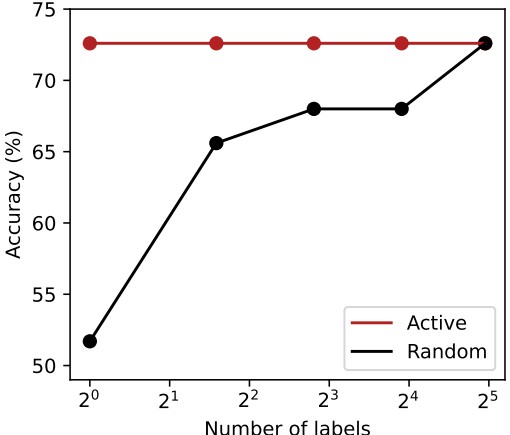

Figure 11: CIFAR label accuracy of chosen head in DivDis vs label used for the Disambiguate stage. The active querying strategy chooses the best head with even one label query, and the random query strategy similarly works with a modest budget of 32 queries.

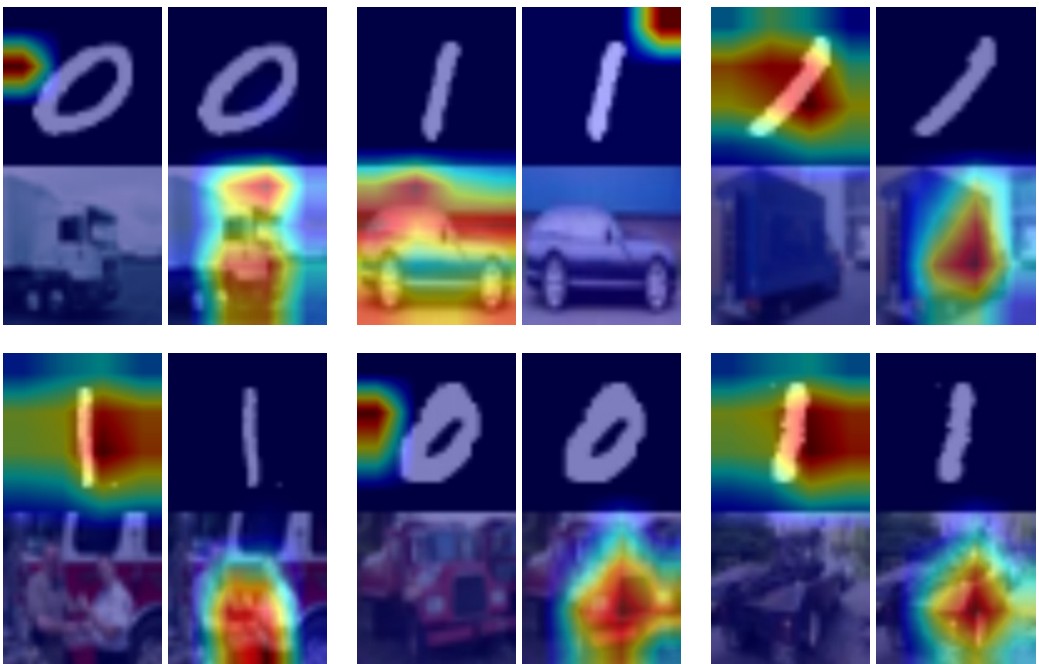

Figure 12: Additional GradCAM visualizations of two learned heads on 6 examples from the source dataset of the MNIST-CIFAR task. These examples sufficiently differentiate the best of the two heads, demonstrating the viability of the source data inspection strategy for the DISAMBIGUATE stage.

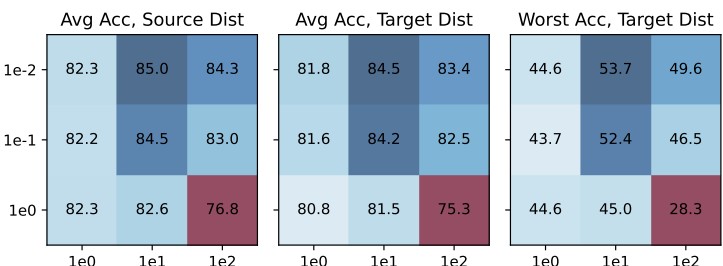

Figure 13: Grids for DivDis's two hyperparameters $(\lambda_1, \lambda_2)$ on the CelebA dataset. Rows indicate $\lambda_1$ and columns indicate $\lambda_2$. We show three metrics measured with held-out datapoints: average accuracy on the source and target distributions and worst-group accuracy on the target distribution. We average each metric across three random seeds. The high correlation between the three metrics indicates that we can tune the hyperparameters of DivDis using only held-out labeled source data.

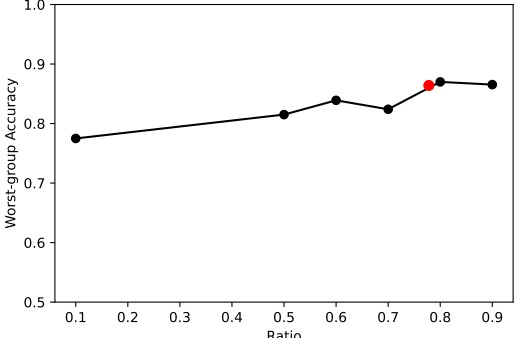

Figure 14: Worst-group accuracy on the Waterbirds benchmark when using different ratio values for $p(y)$ in the regularizer loss (4). The plot shows that the performance of DivDis is not very sensitive to this hyperparameter.

