# OpenReview forum: "Diversify and Disambiguate: Out-of-Distribution Robustness via Disagreement"
_ICLR.cc/2023/Conference — ICLR 2023 poster_

### Official Review · Reviewer_X9wJ · 2022-10-15

**Confidence:** 4
**Correctness:** 4
**Technical Novelty And Significance:** 3
**Empirical Novelty And Significance:** 3
**Recommendation:** 8

**Clarity, Quality, Novelty And Reproducibility:**

- The writing is clear except for a few parts mentioned in the weaknesses section.
- Since the research direction itself seems novel, the proposed approach is also novel.
- The code is publicly available.

**Strength And Weaknesses:**

**Strengths (S):**

[S1] The paper addresses a very important issue where the training data alone may not uniquely specify the best solution for the target distribution. This is a real issue for the current DNNs, since they are limited to producing a single hypothesis, which may or may not be the best way to perform the task. The work provides a principled way to uncover multiple plausible hypotheses and select the best one given a target distribution.

[S2] The process of training diverse heads on unlabeled target distribution is logical. The experiments with the toy dataset and completely/partially correlated spurious factors clearly demonstrate the ability to learn different functions and the ability to choose the best one. However, I think the paper needs to clarify the points mentioned in the weaknesses section.

[S3] The results showing that the hyperparameters that work on held-out source data work for target distribution too is important. This eases hyperparameter tuning.

**Weaknesses (W):**

 [W1] Is it realistic to expect to have access to unlabeled data that has the SAME distribution as the test data? Wouldn't it be more realistic that the unlabeled data is distributed different from both train and test? How would the approach handle that situation?

[W2] For each dataset/experiment, I think the paper should clarify how the unlabeled, target distribution was constructed. Where do the samples come from (a separate validation set?)? Does the size of unlabeled target set matter to ensure diversity?

[W2.1] For Camelyon17, are both unlabeled data and test data from the same hospital?

[W3] Most experiments are limited to 2 heads, so it is unclear how well the method generalizes beyond that (for non-toy datasets).

[W4] Moreover, the separate head approach assumes prior knowledge of the number of plausible hypotheses. I wonder what happens when there is a mismatch between the actual number of plausible hypotheses for the task and the number of heads in the model (and not just toy dataset).

[W5] For the sake of completeness, I think the paper should include methods (JTT/gDRO) for MNIST-CIFAR data with various mixing ratios.


**Summary Of The Paper:**

DivDis tackles the problem of underspecification by generating diverse hypotheses and then disambiguating to choose the best one. Diversification is performed by training multiple heads to yield different predictions on an unlabeled target distribution. Disambiguation is performed by first actively querying to obtain labels for a small target set and then selecting the head with the best accuracy on this set.

**Summary Of The Review:**

Overall, I vote to accept the paper since it addresses an important shortcoming of most DNNs: the inability to consider multiple, plausible ways of performing the task, especially in the absence of a clear signal from the training set. The proposed approach is sound and logical.

Clarifying some of the points I mentioned in the weaknesses could strengthen the paper.

---

> ### Author Response · Authors · 2022-11-13
> **Response to Reviewer X9wJ**
>
> Thank you for your thoughtful comments. We believe your review has helped to significantly improve the paper.
>
> > the separate head approach assumes prior knowledge of the number of plausible hypotheses. I wonder what happens when there is a mismatch between the actual number of plausible hypotheses for the task and the number of heads in the model (and not just toy dataset).
>
> Thanks for raising this point. We evaluated on the collage dataset [1], which necessarily requires at least N=4 heads to cover all four possibilities. Due to space constraints, we added the following new experiments to Appendix A. We will move it into the main text in the final version. We evaluated DivDis with 4 heads:
>
> | Methods | # params | MNIST | SVHN | FMNIST | CIFAR |
> | ----------- | ----------- | ----------- | ----------- | ----------- | ----------- |
> | Upper bound oracle | ----------- | 99.7 | 89.7 | 77.4 | 68.7 |
> | Evading (N=8) [1] | 27152 | 97.3 | 82.1 | 59.6 | 55.8 |
> | Evading (N=16) [1] | 54304 | 96.6 | 72.1 | 64.6 | **58.4** |
> | DivDis (N=2) | 3428 | 98.1 | 66.5 | 62.5 | 51.5 |
> | DivDis (N=4) | 3496 | 99.4 | **75.1** | **66.1** | 53.4 |
>
> This task is constructed so that there are exactly four hypotheses, and we see that using fewer heads (N<4) results in subpar performance. At the same time, DivDis with N=4 shows competitive performance with "Evading" [1] with up to 16 functions. This result highlights the importance of the number of heads in DivDis. Finally, we note that N can be treated as a hyperparameter, and trying out a few small numbers, for example, N=1,2,4, would have sufficed for the datasets in our paper.
>
> In datasets other than collages (Waterbirds, CelebA, MultiNLI, Camelyon), we found that using N=2 suffices and using more heads does not improve performance. Our interpretation is that while these datasets are even more high-dimensional than the collages data, the underlying ambiguity is small enough that N=2 heads can provide sufficient coverage. We also note that the original submission did already include experiments with more heads on the synthetic 2D experiment: Fig 8 uses N=2,3,5 heads, and Fig 2 (right) uses N=20.
>
> [1] Teney, Damien, et al. "Evading the simplicity bias: Training a diverse set of models discovers solutions with superior ood generalization." In CVPR, 2022.
>
> > I think the paper should include methods (JTT/gDRO) for MNIST-CIFAR data with various mixing ratios.
>
> Thank you for the suggestion; we have added JTT and gDRO as additional points of comparison in the MNIST-CIFAR experiments in Fig 5(a). We reproduce the results in the table below:
>
> | Methods | Ratio = 0.0 | Ratio = 0.2 | Ratio = 0.4 | Ratio = 0.6 | Ratio = 0.8 | Ratio = 1.0 |
> | ----------- | ----------- | ----------- | ----------- | ----------- | ----------- | ----------- |
> | ERM | 49.7 | 56.2 | 75.2 | 82.2 | 84.5 | 85.7 |
> | JTT | 50.6 | 51.1 | 75.9 | 83.4 | 85.1 | 85.5 |
> | Group DRO | 49.1 | 62.8 | 80.2 | 84.2 | **86.1** | 86.1 |
> | DivDis | **76.3** | **79.5** | **84.3** | **85.3** | 85.4 | **86.3** |
>
> Both JTT and GDRO fail to work in the challenging complete correlation setting (Ratio = 0.0). As expected, both methods show benefits over ERM in conditions of milder underspecification (0 < mix ratio < 1).
>
> > [W1] Is it realistic to expect to have access to unlabeled data that has the SAME distribution as the test data? Wouldn't it be more realistic that the unlabeled data is distributed different from both train and test? How would the approach handle that situation?
>
> The Camelyon17 experiment uses the "val_unlabeled" split from the U-WILDS benchmark, which follows a different distribution (i.e. a different hospital) from the test data. Thus, DivDis does not require the distribution to be exactly the same as the target distribution – it only requires it to change from the training distribution along a similar axis as the test distribution. We also note that access to unlabeled data from the test distribution is a common assumption in prior works on transductive learning and unsupervised domain adaptation.
>
> > For each dataset/experiment, I think the paper should clarify how the unlabeled, target distribution was constructed. Where do the samples come from (a separate validation set?)?
>
> The unlabeled data for the synthetic 2D experiments is a separate validation set sampled from the target distribution. For MNIST-CIFAR, Waterbirds, CelebA, and MultiNLI, the unlabeled data comes from a separate validation set that follows the target distribution. For CXR, the unlabeled data is a subset of validation data sampled so that the ratio of drain to no drain is 1:1. For Camelyon, the unlabeled data is the official "val_unlabeled" set. We have revised the experiments section to make this clear.

---

> > ### Comment · Reviewer_X9wJ · 2022-11-14
> > **Voting for Acceptance**
> >
> > Thank you authors for addressing the points mentioned in the review.
> >
> > I think the additional results/clarifications will help the paper. I vote for acceptance.

---

### Official Review · Reviewer_kpaT · 2022-10-16

**Confidence:** 5
**Correctness:** 4
**Technical Novelty And Significance:** 3
**Empirical Novelty And Significance:** 3
**Recommendation:** 8

**Clarity, Quality, Novelty And Reproducibility:**

The paper is very clear and well written. I feel like all details are included to replicate this method.

**Strength And Weaknesses:**

### Strengths
- The paper studies a problem of great importance (OOD generalization) that is receiving increasing attention.
- The paper proposes a solution that is complementary to other contributions proposed very recently (appropriated discussed by the authors). This paper encourages diversity across models in prediction space, whereas other solutions work in the space of features or input gradients.
- The disambiguation stage is cleverly addressed with a variety of strategies, which allow comparing a complete method (i.e. including model selection, which some competing works do not address).
- Experiments are performed on a variety of tasks and datasets.

### Weaknesses/questions
- W1: In the conclusion: "An appealing property of DivDis is its automatic discovery of disentangled features, as demonstrated in Sec. 4." Which results does this refer to specifically?

- W2: All experiments (except one on toy data) seem to use 2 heads. Is that right? What happened if the method is applied with >2 heads on the other datasets? The results in (Teney et al. 2021) seem to indicate that more heads are often better. The method in the two papers are different, but it would be interesting to investigate if/why (not) more heads could also help here.

- W3: This paper includes a comparison with the method of (Teney et al. 2021) only on Waterbirds-CC, which seems unfair. I would have liked to see a comparison with the same CIFAR-MNIST (with 4 tiles) used by these other authors, since this dataset can only work with >= 4 heads.

**Summary Of The Paper:**

This paper proposes a solution to underspecification. The authors show how to learn multiple models/hypotheses compatible with the training data (low source risk). They then propose multiple options to select the best one (using additional data such as OOD annotations).

**Summary Of The Review:**

The paper studies an important problem, proposes an innovative solution, and demonstrate very interesting results on a range of datasets. I recommend it for acceptance.

---

> ### Author Response · Authors · 2022-11-13
> **Response to Reviewer kpaT**
>
> Thank you for your thoughtful comments. We believe your review has helped to significantly improve the paper.
>
> > (W2) What happened if the method is applied with >2 heads on the other datasets? The results in (Teney et al. 2021) seem to indicate that more heads are often better.
>
> > (W3) I would have liked to see a comparison with the same CIFAR-MNIST (with 4 tiles) used by these other authors, since this dataset can only work with >= 4 heads.
>
> Thanks for suggesting the collages dataset, which necessarily requires at least N=4 heads to cover all four possibilities. Due to space constraints, we added the following new experiments to Appendix A. We will move it into the main text in the final version. We evaluated DivDis with 4 heads:
>
> | Methods | # params | MNIST | SVHN | FMNIST | CIFAR |
> | ----------- | ----------- | ----------- | ----------- | ----------- | ----------- |
> | Upper bound oracle | ----------- | 99.7 | 89.7 | 77.4 | 68.7 |
> | Evading (N=8) [1] | 27152 | 97.3 | 82.1 | 59.6 | 55.8 |
> | Evading (N=16) [1] | 54304 | 96.6 | 72.1 | 64.6 | **58.4** |
> | Evading + weight sharing (N=16) [1] | 3904 | 99.7 | 50.8 | 50.3 | 50.2 |
> | Evading + weight sharing (N=32) [1] | 4448 | 99.6 | 50.7 | 50.1 | 50.2 |
> | DivDis (N=2) | 3428 | 98.1 | 66.5 | 62.5 | 51.5 |
> | DivDis (N=4) | 3496 | 99.4 | **75.1** | **66.1** | 53.4 |
>
> As expected, DivDis with N=2 shows subpar performance while DivDis with N=4 shows competitive performance with "Evading" [1] with up to 16 functions. We see that DivDis requires substantially fewer functions to achieve the same performance. Furthermore, DivDis uses different linear heads, whereas Evading uses N separate networks. Applying the same weight sharing to the Evading method caused a substantial drop in performance.
>
> We further note that the original collages dataset uses quite small images (16x16); see figure 7 in the appendix for visualizations. We evaluated Evading and DivDis on larger images (32x32):
>
> | Methods | MNIST | SVHN | FMNIST | CIFAR |
> | ----------- | ----------- | ----------- | ----------- | ----------- |
> | Evading (N=8) [1] | 97.3 | 82.1 | 59.6 | 55.8 |
> | Evading (N=16) [1] | 96.6 | 72.1 | 64.6 | 58.4 |
> | Evading (N=8, ours) [1] | 99.7 | 70.0 | 56.3 | 50.9 |
> | Evading (N=16, ours) [1] | 99.7 | 86.3 | 65.9 | 60.8 |
> | ----------- | ----------- | ----------- | ----------- | ----------- |
> | Evading (N=8, x2 images) [1] | 99.5 | 49.7 | 51.6 | 49.7 |
> | Evading (N=16, x2 images) [1] | 99.8 | 49.6 | 52.0 | 49.7 |
> | DivDis (N=2, x2 images) | 97.6 | 63.9 | 61.8 | 51.7 |
> | DivDis (N=4, x2 images) | 96.5 | **73.0** | **69.2** | **52.7** |
>
> These results indicate that input-space diversification methods like Evading are sensitive to input dimension, whereas DivDis is an output-space diversification method and scales better to larger images. We further note that such scalability is also reflected in results on the large-scale Camelyon dataset: DivDis achieves 90.4% accuracy on test hospitals, whereas [2] achieves 82.5%.
>
> In datasets other than collages (Waterbirds, CelebA, MultiNLI, Camelyon), we found that using N=2 suffices and using more heads does not improve performance. Our interpretation is that while these datasets are even more high-dimensional than the collages data, the underlying ambiguity is small enough that N=2 heads can provide sufficient coverage. We also note that the original submission did already include experiments with more heads on the synthetic 2D experiment: Fig 8 uses N=2,3,5 heads, and Fig 2 (right) uses N=20.
>
> [1] Teney, Damien, et al. "Evading the simplicity bias: Training a diverse set of models discovers solutions with superior ood generalization." In CVPR, 2022.
>
> [2] Teney, Damien, Maxime Peyrard, and Ehsan Abbasnejad. "Predicting is not understanding: Recognizing and addressing underspecification in machine learning." In ECCV, 2022
>
>
>
> > W1: In the conclusion: "An appealing property of DivDis is its automatic discovery of disentangled features, as demonstrated in Sec. 4." Which results does this refer to specifically?
>
> We refer to Figure 6 and Table 2 in section 4.3, where we show that, unlike previous methods, DivDis can achieve high worst-group accuracy in the setting where group labels are not used during hyperparameter tuning. This indicates that, to some extent, the two heads are each learning the two features present in the dataset: e.g., bird features and background features in Waterbirds. This happens without any direct supervision of the two different features. Our use of the term "disentanglement" is informal: the features don't necessarily satisfy any formal axis-alignment conditions.

---

> > ### Comment · Reviewer_kpaT · 2022-11-13
> > **Happy with improvements, recommends acceptance**
> >
> > Thanks a lot to the authors for yet improving the paper. I strongly recommend acceptance.
> >
> > > "An appealing property of DivDis is its automatic discovery of disentangled features" (...) Our use of the term "disentanglement" is informal: the features don't necessarily satisfy any formal axis-alignment conditions
> >
> > I would suggest rewording to avoid an informal use of a term ("disentanglement") that some recent works have tried hard to make it more formal. How about "automatic discovery of meaningful complementary features"? The gain in clarity should be in the authors' interest, but this is just a suggestion. No need to reply.

---

> > > ### Author Response · Authors · 2022-11-18
> > > **Thanks! Agreed on disentanglement.**
> > >
> > > Thank you for the specific suggestion! We agree; this new term is closer to what we mean (and what we have evidence for). We have edited the conclusion to weaken our wording around disentangled features.

---

### Official Review · Reviewer_5h8Y · 2022-10-21

**Confidence:** 4
**Correctness:** 4
**Technical Novelty And Significance:** 3
**Empirical Novelty And Significance:** 3
**Recommendation:** 8

**Clarity, Quality, Novelty And Reproducibility:**

The paper is very clear. Nice justifications and motivations are provided.
The contributions are fairly novel compared to concurrent approaches.
A code snippet is provided for one aspect of the method. I haven't seen some information about a repo where the entire code can be found. The method is simply enough to be reproduced event without code sharing, but I recommend to share the code anyway.

**Strength And Weaknesses:**

Pros :
- The paper is very clear and easy to understand.
- The contributions are well framed and positioned
- The contributions are supported by clear experimental evidence

Cons :
- The methods required unveiling the class labels of a selection of test points.
- The approach is limited to underspecified data otherwise diverse predictors cannot be trained on the source distribution.


**Summary Of The Paper:**

This paper adresses distribution shift between train and test samples. The authors leverage the fact that different models can perform equally on the train distribution while exhibiting diversity in their prediction on the test distribution. The diversity among trained model is enforced by explicitly requesting different predictions on unlabeled test data. More precisely, this is done by adding a loss term that minimizes mutual information between pairs of predictions from two models during their respective trainings. Another loss term avoids learning naive classifiers such as constant functions.
At inference time, one needs to select the appropriate model. This is done by querying the label of few data points for which disagreement among trained models is high and examining average accuracy on these points.



**Summary Of The Review:**

I really enjoyed reading this paper and I have only a few remarks.

How does the model scale to large number of heads ? I suppose the MI term will be required for each pair of heads. Could this be a bottleneck ?

I was expecting a bit more experiments of the number of necessary test point label reveals necessary. If possible, it would be nice to extend this.

Since the methods is useful for underspecified data how can we assess to match this use case when start with a new dataset ?

---

> ### Author Response · Authors · 2022-11-13
> **Response to Reviewer 5h8Y**
>
> Thank you for your thoughtful comments. We believe your review has helped to significantly improve the paper.
>
> > How to assess the viability of DivDis given a new dataset
>
> This is an interesting question, which we believe will become even more important as ML systems are deployed in real-world distribution shift conditions. The problem of detecting distribution shift and determining which method(s) to use is an open problem in the robustness literature. As rough guidelines, we suggest first starting with a single model to establish a baseline for performance. For DivDis, N=2 heads seem to suffice for multiple real datasets, so we recommend starting with N=2 and the default hyperparameters in our experiments. DivDis may be useful as a way to detect underspecification, e.g., by testing if it can learn multiple distinct solutions, but we leave the investigation of this to future work.
>
> > How does the model scale to large number of heads? I suppose the MI term will be required for each pair of heads. Could this be a bottleneck?
>
> To train DivDis with multiple heads, we calculate each pairwise mutual information term. While the computation scales as O(N^2) with the number of heads N, the time and memory required are low compared to feeding forward through the network backbone. Also, the MI objective is easily parallelizable: see Appendix A. Therefore, for moderate values of N (up to at least 100), MI term calculation is not a bottleneck.
>
> > I recommend to share the code
>
> In the final version, we will include a link to a public repository as a footnote on page 1.

---

### Official Review · Reviewer_jJSw · 2022-10-25

**Confidence:** 4
**Correctness:** 3
**Technical Novelty And Significance:** 3
**Empirical Novelty And Significance:** 3
**Recommendation:** 6

**Clarity, Quality, Novelty And Reproducibility:**

--- **Post Rebuttal** ---
- This work has ***good clarity***: it clearly presents the problem, method, related works, and experiment.
- The proposed method (DivDis) is reasonable to me. The idea of learning multiple functions is already explored and studied for the same problem of underspecification. However, the new contributions based on this idea are convincing.
- This work gives ***sufficient details for the reproducibility***

**Strength And Weaknesses:**

**[Strength]**

+ This work is well written. The problem, method, and experiment are clearly presented.

+ The related work is nicely organized.

+ The proposed DivDis is straightforward and has a clear motivation. Learning multiple and diverse functions on the underspecified dataset is reasonable. Then, selecting the most suitable function is also straightforward.

+ The three strategies in stage two of DivDis are good. This work clearly presents them. Active querying required as little as a single label is interesting to me. In addition, this work highlights in Section 3.3 that "We emphasize that existing OOD methods tune hyperparameters using target set labels" Please clarify how the existing methods use target labels and give some references or examples.

**[Weakness]**

- While the framework is reasonable, its novelty is somewhat weak. First, existing methods already study the idea of learning diverse functions. For example, works [1,2] also study the underspecification issue and propose to learn multiple different functions. From this point, the novelty of this work is limited.

- Following up on the above, the first stage of this work (i.e., diversity) uses a mutual information loss to learn diverse functions. In comparison, [1] uses a penalty on the alignment of their input gradients; [2] uses a local independence loss and an on-manifold loss. Then, a question is: which manner is more effective to learn multiple diverse functions? A discussion or experimental analysis would be helpful.

- The comparison with [1,2] should be clearly reported in the experiment. I noticed that methods of [1,2] do not have a simple function selection while this work has (stage 2). So, this work can use the same selection manner of [1,2] for comparison.

    [1] Teney, Damien, et al. "Evading the simplicity bias: Training a diverse set of models discovers solutions with superior ood generalization." In CVPR, 2022.

    [2] Teney, Damien, Maxime Peyrard, and Ehsan Abbasnejad. "Predicting is not understanding: Recognizing and addressing underspecification in machine learning." In ECCV, 2022

 - Crucial details are missing: 1) according to Section B.2, this work only uses N=2 heads. Namely, this work only learns two functions. Why use N=2? What is the effect of N on learning? 2) when the dataset is more complex (e.g., collages dataset of [2]), using N=2 might not suitable. Thus, this work needs to study N.

-  In domain adaptation, using the disagreement of two classifiers is beneficial for achieving high accuracy on the target data [3]. Table 3 compares several domain adaptation methods. It would be better if this work could include [3] in the table.

--- **Post Rebuttal** ---
The authors have provided responses for the above questions. The rebuttal is helpful and addresses most of my concerns.


**Summary Of The Paper:**

This paper considers the problem of ***underspecification*** of the dataset. To address this issue, this paper proposes a *two-stage* framework, Diversify and Disambuguate (DivDis), for learning from underspecified data. For the first stage (***I. diversity***), DivDis learns a diverse set of hypotheses that gain low source loss but make different predictions on target data. Then, in the second stage (***II. disambiguate***), DivDis aims to select one of the learned functions. Note that, the work uses additional information for the function selection. In the experiment, this work validates the effectiveness of DivDis on datasets with *a complete correlation* (the source distribution has a spuriously correlated attribute that can predict the label perfect accuracy).



**Summary Of The Review:**

--- **Post Rebuttal** ---

The proposed method (DivDis) is reasonable to me. The idea of learning multiple functions is already explored and studied for the same problem of underspecification. However, ***the new contributions based on this idea are convincing and sufficient***. The authors also discuss the effect of the number of heads (N), which corresponds to the number of functions. Therefore, I tend to accept this paper.

---

> ### Author Response · Authors · 2022-11-13
> **Response to Reviewer jJSw (1/2)**
>
> Thank you for your thoughtful comments. We believe your review has helped to significantly improve the paper.
>
> > this work only uses N=2 heads…when the dataset is more complex (e.g., collages dataset of [2]), using N=2 might not be suitable.
> > A clear comparison with existing methods (especially [1-2] in the weakness above) is needed.
>
> Thanks for suggesting the collages dataset, which necessarily requires at least N=4 heads to cover all four possibilities. Due to space constraints, we added the following new experiments to Appendix A. We will move it into the main text in the final version. We evaluated DivDis with 4 heads:
>
> | Methods | # params | MNIST | SVHN | FMNIST | CIFAR |
> | ----------- | ----------- | ----------- | ----------- | ----------- | ----------- |
> | Upper bound oracle | ----------- | 99.7 | 89.7 | 77.4 | 68.7 |
> | Evading (N=8) [1] | 27152 | 97.3 | 82.1 | 59.6 | 55.8 |
> | Evading (N=16) [1] | 54304 | 96.6 | 72.1 | 64.6 | **58.4** |
> | Evading + weight sharing (N=16) [1] | 3904 | 99.7 | 50.8 | 50.3 | 50.2 |
> | Evading + weight sharing (N=32) [1] | 4448 | 99.6 | 50.7 | 50.1 | 50.2 |
> | DivDis (N=2) | 3428 | 98.1 | 66.5 | 62.5 | 51.5 |
> | DivDis (N=4) | 3496 | 99.4 | **75.1** | **66.1** | 53.4 |
>
> As expected, DivDis with N=2 shows subpar performance while DivDis with N=4 shows competitive performance with "Evading" [1] with up to 16 functions. We see that DivDis requires substantially fewer functions to achieve the same performance. Furthermore, DivDis uses different linear heads, whereas Evading uses N separate networks. Applying the same weight sharing to the Evading method caused a substantial drop in performance.
>
> We further note that the original collages dataset uses quite small images (16x16); see figure 7 in the appendix for visualizations. We evaluated Evading and DivDis on larger images (32x32):
>
> | Methods | MNIST | SVHN | FMNIST | CIFAR |
> | ----------- | ----------- | ----------- | ----------- | ----------- |
> | Evading (N=8) [1] | 97.3 | 82.1 | 59.6 | 55.8 |
> | Evading (N=16) [1] | 96.6 | 72.1 | 64.6 | 58.4 |
> | Evading (N=8, ours) [1] | 99.7 | 70.0 | 56.3 | 50.9 |
> | Evading (N=16, ours) [1] | 99.7 | 86.3 | 65.9 | 60.8 |
> | ----------- | ----------- | ----------- | ----------- | ----------- |
> | Evading (N=8, x2 images) [1] | 99.5 | 49.7 | 51.6 | 49.7 |
> | Evading (N=16, x2 images) [1] | 99.8 | 49.6 | 52.0 | 49.7 |
> | DivDis (N=2, x2 images) | 97.6 | 63.9 | 61.8 | 51.7 |
> | DivDis (N=4, x2 images) | 96.5 | **73.0** | **69.2** | **52.7** |
>
> These results indicate that input-space diversification methods like Evading are sensitive to input dimension, whereas DivDis is an output-space diversification method and scales better to larger images. We further note that such scalability is also reflected in results on the large-scale Camelyon dataset: DivDis achieves 90.4% accuracy on test hospitals, whereas [2] achieves 82.5%.
>
> In datasets other than collages (Waterbirds, CelebA, MultiNLI, Camelyon), we found that using N=2 suffices and using more heads does not improve performance. Our interpretation is that while these datasets are even more high-dimensional than the collages data, the underlying ambiguity is small enough that N=2 heads can provide sufficient coverage. We also note that the original submission did already include experiments with more heads on the synthetic 2D experiment: Fig 8 uses N=2,3,5 heads, and Fig 2 (right) uses N=20.
>
> [1] Teney, Damien, et al. "Evading the simplicity bias: Training a diverse set of models discovers solutions with superior ood generalization." In CVPR, 2022.
>
> [2] Teney, Damien, Maxime Peyrard, and Ehsan Abbasnejad. "Predicting is not understanding: Recognizing and addressing underspecification in machine learning." In ECCV, 2022
>
> > Novelty is somewhat weak…existing methods already study the idea of learning diverse functions.
>
> Indeed, prior works, including [1,2], have noted that diversity is effective in underspecified settings. As stated in our related work section, diversity is emphasized in the literature for ensembles, Rashomon sets, maximum entropy, and quality diversity. Our core novelty lies in our mutual information loss, enabling output-space diversification using unlabeled OOD data. In contrast, both papers [1,2] diversify using input-space gradients, requiring different solutions corresponding to different input regions. Additionally, our "Disambiguate" stage explicitly considers the step of choosing a single function to deploy given a diverse set; this step is important for the real-world deployment of such methods and is not considered in [1,2]. We have edited the text to clarify these points and added the experiment above.

---

> > ### Author Response · Authors · 2022-11-13
> > **Response to Reviewer jJSw (2/2)**
> >
> > > this work highlights in Section 3.3 that "We emphasize that existing OOD methods tune hyperparameters using target set labels" Please clarify how the existing methods use target labels and give some references or examples.
> >
> > For hyperparameter tuning, existing methods such as [3,4,5] use a labeled validation set from the target distribution. Our active and random querying strategies use a small subset (less than 1%) of this labeled validation set. Therefore, the active and random query strategies do not use any information unavailable to previous approaches. We have clarified this in the main text.
> >
> > [3] Levy, Daniel, et al. "Large-scale methods for distributionally robust optimization." Advances in Neural Information Processing Systems 33 (2020): 8847-8860.
> >
> > [4] Nam, Junhyun, et al. "Learning from failure: De-biasing classifier from biased classifier." Advances in Neural Information Processing Systems 33 (2020): 20673-20684.
> >
> > [5] Liu, Evan Z., et al. "Just train twice: Improving group robustness without training group information." International Conference on Machine Learning. PMLR, 2021.

---

> > ### Comment · Reviewer_jJSw · 2022-11-16
> > **Good Rebuttal**
> >
> > Thank the authors for providing the rebuttal!
> >
> > - Helpful discussion on the number of heads (N).
> > - Convincing clarification of new contributions (although diversity is not a new idea).
> >
> > Therefore, I increase my score and tend to accept this paper.

---

### Author Response · Authors · 2022-11-13
**General Response to All Reviewers and Summary of Revisions**

We thank all reviewers for their thoughtful comments. We believe that the feedback has helped improve the paper.

Based on the feedback, we ran additional experimental comparisons and uploaded a revised pdf with modifications in blue. The major changes are:
- Added experiments on the 4-way Collages dataset from Teney et al., in which we evaluate DivDis with more than two heads (jJSw, kpaT, X9wJ, Appendix A, Tables 4-5, Figure 7).
- Added JTT and Group DRO as additional points of comparison for the CIFAR-MNIST experiment (X9wJ, Fig 5(a))
- Edited the paper to more clearly state label requirements for the Dis stage (jJSw, page 5) and specify what the unlabeled set is for each setting (X9wJ, pages 6-8)

We address specific concerns in individual replies to each reviewer. If there are any remaining questions or concerns, please let us know!

---

### Decision · Program_Chairs · 2023-01-20

**Decision:**

Accept: poster

**Justification For Why Not Higher Score:**

The strengths of the paper are that it pushes the boundaries of out-of-distribution generalization to problems with ambiguity at training time. The results are also relatively strong.

**Justification For Why Not Lower Score:**

The weaknesses of the paper are that the notion of diversity was statistical independence, which was justified in an adhoc manner and that the disambiguation can be haphazard if the test set is not different enough.

**Metareview: Summary, Strengths And Weaknesses:**

This paper develops an algorithm for out-of-distribution generalization when the training distribution does not ground the behavior at test time. The approach works by constructing a diverse set of predictors measure by information, then uses some extra labels to disambiguate and choose among the diverse set of predictors.

The strengths of the paper are that it pushes the boundaries of out-of-distribution generalization to problems with ambiguity at training time. The results are also relatively strong.

The weaknesses of the paper are that the notion of diversity was statistical independence, which was justified in an adhoc manner and that the disambiguation can be haphazard if the test set is not different enough.

The reviewers were already on average positive. There were questions about the contribution of diversity and the choice of the number of heads from one reviewer. Both of which were resolved in the author reply. There was also a discussion on the use of the word "disentanglement," which already has a strong meaning in machine learning. This was acknowledged and addressed as well.

The paper should be accepted because it makes a simple and clear contribution to pushing the boundary of out of distribution generalization.

**Note From Pc:**

if the above contains the word "oral" or "spotlight" please see: "oral" presentation means -> notable-top-5% and "spotlight" means -> notable-top-25%. As stated in our emails, we are disassociating presentation type from AC recommendations